# T-follicular helper cell profiles differ by malaria antigen and for children compared to adults

Catherine Suzanne Forconi[1], Christina Nixon[2], Hannah W Wu[2], Boaz Odwar[3], Sunthorn Pond-Tor[2], John M Ong'echa[3], Jonathan D Kurtis[2], Ann M Moormann[1]*

[1]Division of Infectious Diseases and Immunology, Department of Medicine, University of Massachusetts Chan Medical School, Worcester, United States; [2]Department of Pathology and Laboratory Medicine, Brown University, Providence, United States; [3]Center for Global Health Research, Kenya Medical Research Institute, Kisumu, Kenya

*For correspondence:
ann.moormann@umassmed.edu

## eLife Assessment

This descriptive study used multiparameter spectral flow cytometry and clustering analysis of a subset of CD4 T cells, termed circulating T follicular helper (cTfh), responding to *Plasmodium falciparum* antigens, PfSEA -1A and PfGARP. The results from this comprehensive study provide **valuable** information regarding differences in cTfh response profiles between children and adults living in malaria-endemic Kenya and thus offer a potential usefulness towards improving choices of antigen candidates for malaria vaccines. However, the analysis and interpretation of antigen-specific CD4 cTfh responses remain **incomplete**.

**Abstract** Circulating T-follicular helper ($cT_{FH}$) cells have the potential to provide an additional correlate of protection against *Plasmodium falciparum* (*Pf*) as they are essential to promote B-cell production of long-lasting antibodies. Assessing the specificity of $cT_{FH}$ subsets to individual malaria antigens is vital to understanding the variation observed in antibody responses and identifying promising malaria vaccine candidates. Using spectral flow cytometry and unbiased clustering analysis, we assessed antigen-specific $cT_{FH}$ cell recall responses in vitro to malaria vaccine candidates *Pf*-schizont egress antigen-1 (*Pf*SEA-1A) and *Pf*-glutamic acid-rich protein (*Pf*GARP) within a cross-section of children and adults living in a malaria-holoendemic region of western Kenya. In children, a broad array of $cT_{FH}$ subsets (defined by cytokine and transcription factor expression) were reactive to both malaria antigens, *Pf*SEA-1A and *Pf*GARP, while adults had a narrow profile centering on $cT_{FH}17$- and $cT_{FH}1/17$-like subsets following stimulation with *Pf*GARP only. Because $T_{FH}17$ cells are involved in the maintenance of memory antibody responses within the context of parasitic infections, our results suggest that *Pf*GARP might generate longer-lived antibody responses compared to *Pf*SEA-1A. These findings have intriguing implications for evaluating malaria vaccine candidates as they highlight the importance of including $cT_{FH}$ profiles when assessing interdependent correlates of protective immunity.

## Introduction

Despite progress made to reduce the global malaria burden, *Plasmodium falciparum* (*Pf*) remains one of the leading causes of mortality among children under 5 years of age (*Geneva: World Health Organization, 2022*). Unfortunately, progress has been impeded by a plateau in malaria control since 2015. Antimalarial drug resistance (*Taylor and Juliano, 2014*; *Aydemir et al., 2018*) and unprecedented

logistical challenges during the COVID-19 pandemic that dramatically impacted the distribution of insecticide-impregnated mosquito nets led to an increase in malaria cases in children under 5 years of age from 17.6 million in 2015 to 19.2 million in 2021 in East and Southern African countries (*Geneva: World Health Organization, 2022*; *Ippolito et al., 2021*), thus, contributing to reinvigorated prioritization of malaria vaccine initiatives.

After 30 years of development, the first malaria vaccine—RTS,S/AS01E—was approved by the World Health Organization in 2021 for use in children residing in malaria-endemic regions. However, the limited efficacy of RTS,S, (*RTS,S Clinical Trials Partnership, 2015*) particularly against severe malaria (*RTS,S Clinical Trials Partnership, 2015*), has motivated the search for additional vaccine candidate antigens, including blood-stage *Pf*-schizont egress antigen-1A (*Pf*SEA-1A) (*Raj et al., 2014*) and *Pf*-glutamic acid-rich protein (*Pf*GARP) (*Raj et al., 2020*). Antibodies against *Pf*SEA-1A correlate with significantly lower parasite densities in Kenyan adults and adolescents and substantially reduce schizont replication in vitro (*Raj et al., 2014*; *Nixon et al., 2017*), whereas *Pf*GARP-specific antibodies kill trophozoite-infected erythrocytes in culture and confer partial protection against *Pf* challenge in vaccinated nonhuman primates (*Raj et al., 2020*). The presence of antibodies against *Pf*GARP was associated with a 2-fold-lower parasite density in Kenyan adults and adolescents (*Raj et al., 2020*). Due to the complexity of the parasites' life cycle, there is general agreement that a multivalent vaccine would be more efficacious (*Holder, 1999*). However, determining which antigens hold the most promise for inclusion in next-generation malaria vaccines remains a challenge.

Many pathogens and vaccines engender protective antibody responses after a single or few exposures (*Amanna et al., 2007*), characterized by the production of long-lived plasma cells and memory B cells (*Tarlinton and Good-Jacobson, 2013*). Affinity maturation takes place in the germinal center (GC), where antigen-specific $CD4^{pos}$ T-follicular helper ($T_{FH}$) cells are required not only to provide cellular (CD40L) and molecular (IL-21, IL-4, and IL-13) signals to trigger B-cell proliferation but also to promote GC maintenance and plasmablast differentiation. The GC is also where higher affinity B cells outcompete B cells with lower affinity for $T_{FH}$ help (*Crotty, 2014*; *Crotty, 2019*). $T_{FH}$ cells are defined by the expression of a combination of markers, starting with chemokine receptor CXCR5, which directs $CD4^{pos}$ T cells from the T-cell zone to engage with follicular B cells (*Breitfeld et al., 2000*; *Crotty, 2011*). Once antigen-experienced $T_{FH}$ cells leave the GC, they become circulating $T_{FH}$ ($cT_{FH}$) cells and correlate with the generation of long-lasting antibody responses. Interestingly, $cT_{FH}$ cells can also come from peripheral $cT_{FH}$ precursor $CCR7^{low}PD1^{high}CXCR5^{pos}$ cells; thus, they also have a GC-independent origin (*He et al., 2013*). Expression profiles of CCR6 and CXCR3 further categorize $cT_{FH}$ into subsets as follows: $cT_{FH}1$ ($CCR6^{neg}CXCR3^{pos}$), $cT_{FH}2$ ($CCR6^{neg}CXCR3^{neg}$), and $cT_{FH}17$ ($CCR6^{pos}CXCR3^{neg}$) (*Schmitt et al., 2014*), whereas the expression of PD-1, ICOS, CD127, and CCR7 define their functional status: quiescent/central memory $cT_{FH}$ ($CCR7^{high}PD-1^{neg}ICOS^{neg}CD127^{pos}$) or activated/effector memory $cT_{FH}$ ($CCR7^{low}PD-1^{pos}ICOS^{pos}CD127^{low/neg}$) (*Schmitt et al., 2014*; *Gong et al., 2019*; *Dunham et al., 2008*). In addition, transcription factors (i.e. Bcl6 and cMAF) and cytokines (i.e. interferon-gamma [IFNγ], IL-4, and IL-21) are crucial to further defining the role of each $cT_{FH}$ subset within the context of their interactions with B cells to promote antibody production (*Bélanger and Crotty, 2016*; *Seth and Craft, 2019*; *Olatunde et al., 2021*; *Andris et al., 2017*). Bcl6 facilitates the production of IL-21 by T cells, which aids B-cell affinity maturation and antibody production (*Nurieva et al., 2009*; *Nurieva and Chung, 2010*; *Liu et al., 2013*). $T_{FH}$ cells also secrete cytokines that align with their subset classifications (but are not limited to them), such as IFNγ ($T_{FH}1$), IL-4/IL-13 ($T_{FH}2$), IL-4/IL-5/IL-13 ($T_{FH}13$), and IL-17 ($T_{FH}17$), that direct antibody isotype class-switching and mediate effector functions (*Crotty, 2014*; *Gong et al., 2019*; *Bélanger and Crotty, 2016*; *Seth and Craft, 2019*; *Olatunde et al., 2021*; *Gowthaman et al., 2019*).

Several studies have characterized $cT_{FH}$ subsets within the context of both adult and childhood *Pf*-malaria infections, as well as in healthy malaria-naïve volunteers after controlled malaria infections (*Chan et al., 2020*). In Mali, where malaria is seasonal, Obeng-Adjei and colleagues showed that $T_H1$-polarized $cT_{FH}$ $PD1^{pos}CXCR5^{pos}CXCR3^{pos}$ cells were preferentially activated in children and less efficient than $CXCR3^{neg}$ $cT_{FH}$ cells in helping autologous B cells to produce antibodies; yet they used US healthy adult $cT_{FH}$ cells for their in vitro assays. Therefore, the authors suggested that promoting $T_H2$-like $CXCR3^{neg}$ $cT_{FH}$ subsets could improve antimalarial vaccine efficacy (*Obeng-Adjei et al., 2015*). Similarly, a study in Papua, Indonesia, where malaria is perennial, found that all $cT_{FH}$ subsets from adults showed higher activation and proliferation status compared to $cT_{FH}$ cells from children (*Oyong et al.,*

*2022*). After in vitro stimulation with infected red blood cells (iRBCs), they found that all $cT_{FH}$ subsets were activated in adults accompanied by IL-4 production, whereas only the $T_H1$-polarized $cT_{FH}$ subset responded in children (*Oyong et al., 2022*). One study in Uganda, where malaria is holoendemic with seasonal peaks, showed that a shift from $T_H2$- to $T_H1$-polarized $cT_{FH}$ subsets occurred during the first 6 years of life and was associated with the development of functional antibodies against *Pf*-malaria, yet appeared to be independent of malaria exposure and this age-associated shift was also observed in a malaria-naïve population (*Chan et al., 2022*). Interestingly, this study also found that a higher proportion of $T_H17$-$cT_{FH}$ cells was associated with a decreased risk of *Pf* infection the following year; however, the authors postulated that this phenomenon could have been driven by the previous exposure to the parasite. The observed higher abundance of $T_H2$-like $cT_{FH}$ cells in children younger than 6 years of age and the age-associated increase in $T_H1$-$cT_{FH}$ cells achieving the same proportion of $T_H1$-$cT_{FH}$ cells by 6 years of age and into adulthood appeared to be independent of malaria exposure. Of note, these studies were limited by the use of a two-dimensional gating strategy to classify $cT_{FH}$ subsets and whole parasite activation conditions, either during infections or in vitro stimulation assays, leaving the malaria-antigen specificity of the different $cT_{FH}$ subset responses undefined. However, these studies highlight progress in the field of evaluating the role of human $T_{FH}$ cells in antimalarial immunity.

Antibody levels appear to be unreliable predictors of malaria vaccine efficacy (*Stanisic and McCall, 2021*). Thus, establishing a combined immune profile to incorporate other surrogates of protection is warranted. Here, we examined the profile of $cT_{FH}$ subsets using multiparameter spectral flow cytometry (*Bentebibel et al., 2015*; *Wei et al., 2015*) against two malaria vaccine candidate antigens (*Pf*SEA-1A [*Raj et al., 2014*] and *Pf*GARP [*Raj et al., 2020*]) in a cross-section of children and adults residing in a malaria-holoendemic region of Western Kenya. Our findings revealed significant differences in $cT_{FH}$ subsets between children and adults, where children had more abundant $cT_{FH}1$-like cells and showed antigen-specific responses from all $cT_{FH}$ types, whereas these responses were limited to $cT_{FH}17$- and 1/17-like cells in adults. Moreover, this study showed that *Pf*GARP triggered Bcl6 expression across $cT_{FH}$ subsets, whereas *Pf*SEA-1A induced more cMAF expression, demonstrating the feasibility of implementing T-cell immune correlates to down-select new malaria candidates.

## Results
### Children had lower anti-PfSEA-1A antibodies compared to adults but similar levels of anti-PfGARP antibodies

This cross-sectional study selected a convenience sample of 7-year-old children and adults with a mean age of 22.67 years (ranging from 19 to 30 years of age). No statistical difference was observed (p=0.35 after Welch's test) regarding the absolute lymphocyte count (ALC) between children (mean of 373.1; standard deviation [SD] of 118.1) and adults (mean of 335.2; SD of 99.08). Children had significantly

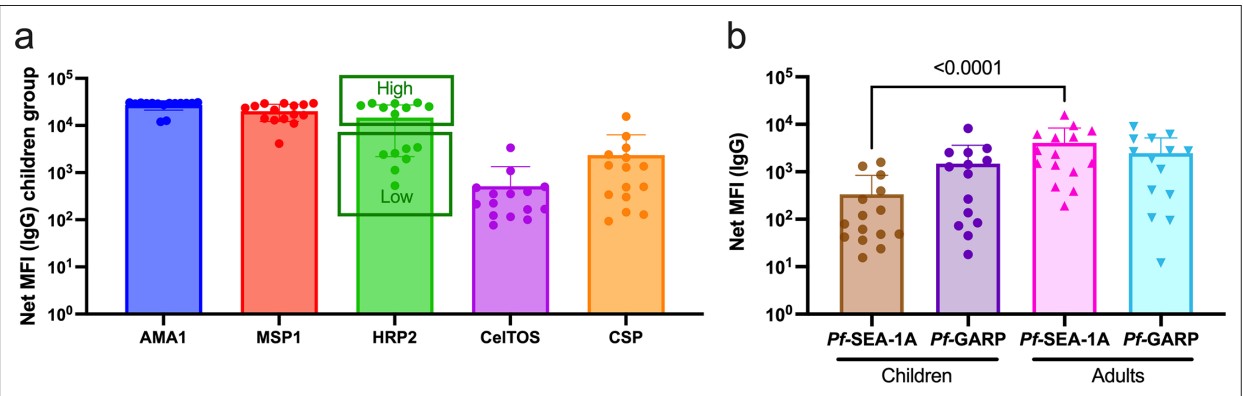

**Figure 1.** *Pf*-malaria IgG sero-profiles for children and adults. (**a**) IgG antibody levels against AMA1, MSP1, HRP2, CelTos, and CSP for children (n=14). The bar plots indicate the mean with standard deviation (SD). (**b**) IgG antibody levels against *Pf*SEA-1A and *Pf*GARP comparing children (n=15) and adults (n=15). The bar plots indicate the mean with SD. Net median fluorescence intensity (MFI) values are the antigen-specific MFI values minus the BSA background. Mann-Whitney tests were performed.

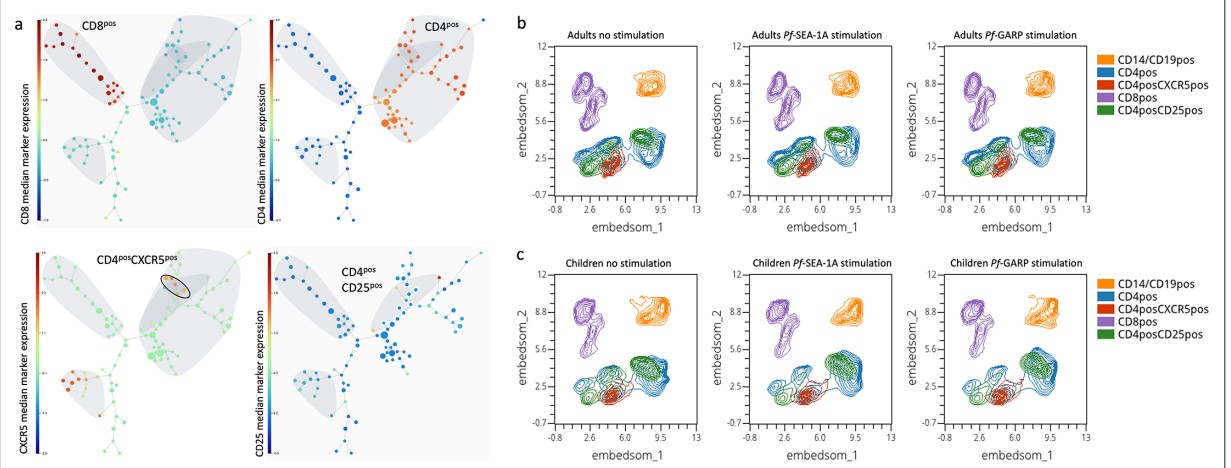

**Figure 2.** Cluster visualization of immune cell types and their abundance. (**a**) A 100-node and 75-meta-cluster FlowSOM tree was generated on live lymphocytes from all our participants (n=29 children and adults), highlighting CD3posCD8pos, CD3posCD4pos, CD4posCXCR5pos, and CD4posCD25pos cells. The five nodes of the CD4posCXCR5posCD25neg population are circled and were used for the downstream analysis. The colored scale was based on the median arcsinh-transformed marker expression. Uniform Manifold Approximation and Projection (UMAP) plots showing five clusters defined as follows: CD14pos and CD19pos (orange), CD4pos (blue), CD4posCXCR5pos (red), CD8pos (purple), and CD4posCD25pos (green) from peripheral blood mononuclear cells (PBMCs) isolated from (**b**) adults (n=15) and (**c**) children (n=14) that were unstimulated, stimulated with PfSEA-1A, or stimulated with PfGARP, respectively, from left to right.

The online version of this article includes the following figure supplement(s) for figure 2:

**Figure supplement 1.** Representative cytoplots of the extracellular flow staining.

**Figure supplement 2.** EdgeR analysis of the CD4pos, CD4posCXCR5pos, CD8pos, and CD14/CD19pos populations under different stimulation conditions in adults (n=15) and children (n=14).

lower hematocrit values (median of 37%, interquartile range [IQR] of 32.7–40.5) compared to adults (median of 44.4%, IQR of 40.5–46), p=0.002 after a two-tailed Mann-Whitney t-test; however, these values were within normal ranges after adjusting for age (*Pluncevic Gligoroska et al., 2019*). Both males and females were enrolled, with 43% (6/14) and 53% (8/15) being female children or adults, respectively. Seroprofiles against a panel of commonly used malaria antigens were generated to confirm the history of previous Pf-infections for the selected children (*Figure 1a*). All participants had high IgG levels against merozoite antigens, apical membrane antigen 1 (AMA-1), and merozoite surface protein (MSP1), confirming at least one Pf-infection within their lifetime (*Yman et al., 2019*; *O'Flaherty et al., 2021*). IgG antibodies against circumsporozoite protein (CSP) and CelTOS (liver-stage antigens) were characteristically lower than against blood-stage antigens, yet were present in all study children. We observed a clear bimodal distribution in antibody levels against histidine-rich protein 2 (HRP2) with half of the children having 'high-HRP2' vs 'low-HRP2' IgG levels, possibly a reflection of recent malaria history since it has been suggested that HRP2-specific antibodies are short-lived and could serve as a surrogate for a recent infection (*Turnbull et al., 2022*). We assessed serological profiles for two Pf-malaria antigens being considered as potential vaccine candidates (*Raj et al., 2014*; *Raj et al., 2020*), PfSEA-1A and PfGARP (*Figure 1b*). We found that children had a significantly lower median level of IgG against PfSEA-1A, compared to adults (p<0.0001), whereas median levels to PfGARP were similarly high for adults and children, yet with a broad range of reactivity.

## The overall abundance of CD4posCXCR5pos cells is unaltered by in vitro antigen stimulation

To determine whether in vitro antigen stimulation with PfSEA-1A or PfGARP altered the abundance of total cTFH cells, we compared cTFH cells from adults and children using a FlowSOM unbiased clustering analysis and EMBEDSOM dimensional reduction based on common lineage markers assessed by spectral flow cytometry from 87,812 live lymphocytes from each sample: CD8pos, CD4pos, CD4posCXCR5pos, and CD4posCD25pos (*Figure 2a*). As expected, after a short stimulation (6 hr), overall abundances of cTFH cells were similar across conditions for both adults (*Figure 2b*) and children (*Figure 2c*).

This observation was confirmed by EdgeR statistical analysis (*Figure 2—figure supplement 2*) and demonstrated that in vitro stimulation did not preferentially expand any of the T-cell populations on which we based our subsequent analyses.

## An unbiased clustering analysis identifies 12 distinct cT$_{FH}$ meta-clusters

Numerous markers and a two-dimensional gating strategy have previously been used to determine the frequency of cT$_{FH}$ subsets (*Obeng-Adjei et al., 2015*; *Oyong et al., 2022*; *Chan et al., 2022*). To simultaneously account for the expression of 17 markers required to define cT$_{FH}$ subsets, we instead used FlowSOM unbiased clustering analysis to determine the frequency of cT$_{FH}$ subsets from a pool

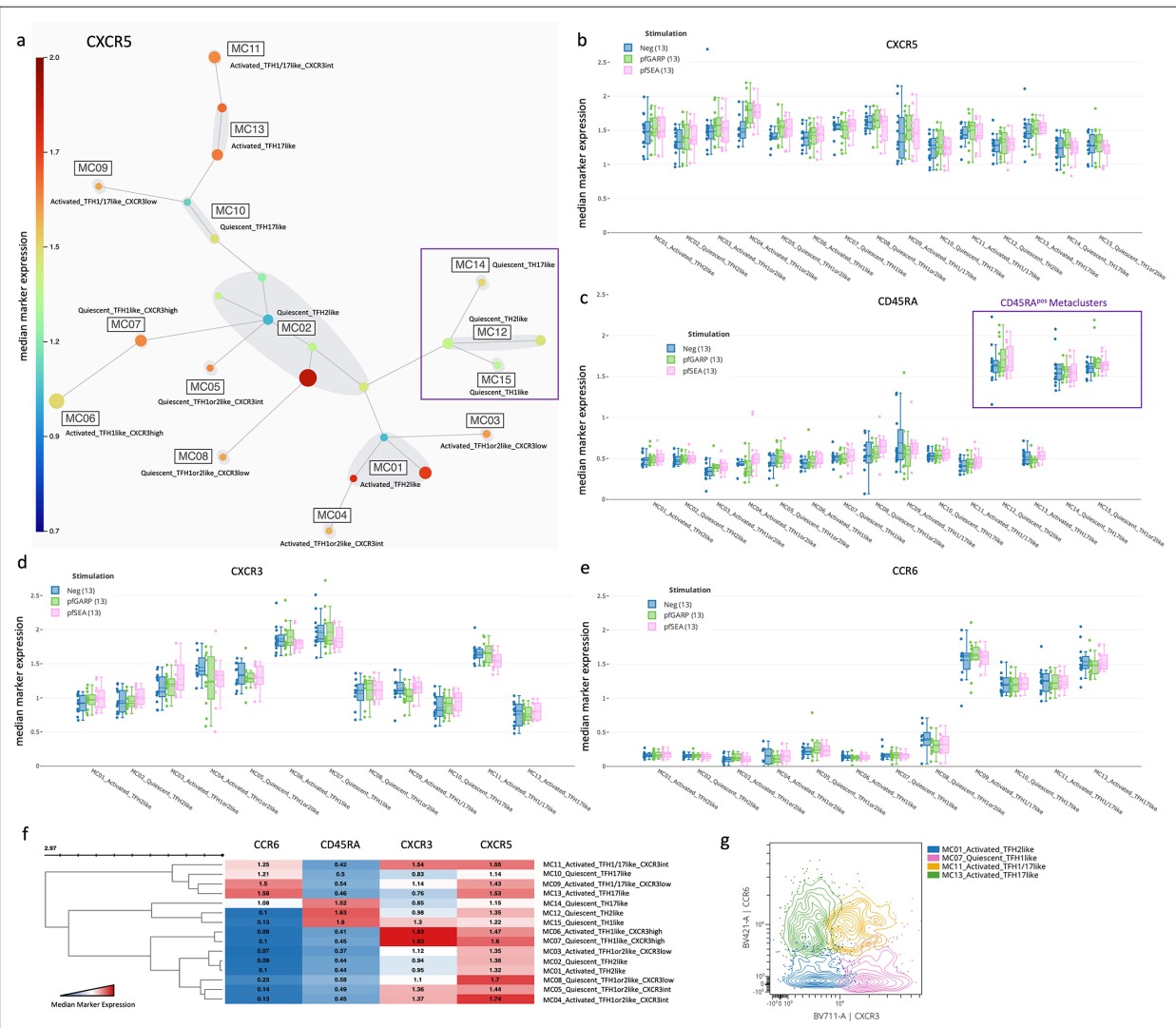

**Figure 3.** Characterization of T$_{FH}$ subsets by CXCR5, CXCR3, CCR6, and CD45RA expression. (**a**) A new 25-node and 15 meta-cluster FlowSOM tree was generated from the five CD4$^{pos}$CXCR5$^{pos}$CD25$^{neg}$ nodes as shown in *Figure 2a* (CXCR5 expression tree). The colored scale of CXCR5 expression was based on the median arcsinh-transformed with red as the highest and deep blue as the absence of CXCR5 expression. (**b**) Box plots showing CXCR5 expression across all cT$_{FH}$-like meta-clusters from children (n=13) with no stimulation (blue) and after in vitro stimulation with *Pf*GARP (green) or *Pf*SEA-1A (pink). Fifty percent of the data points are within the box limits, the solid line indicates the median, the dashed line indicates the mean, and the whiskers indicate the range of the remaining data with outliers being outside that range. Similar box plots are shown for (**c**) CD45RA, (**d**) CXCR3, and (**e**) CCR6 expression across all cT$_{FH}$-like meta-clusters. (**f**) Clustered heatmap showing the median arcsinh-transformed expression for CCR6, CD45RA, CXCR3, and CXCR5 across meta-clusters, red showing the highest expression and blue the lowest. (**g**) Cytoplots of CCR6 vs CXCR3 expression where MC01 is blue, MC07 is pink, MC11 is yellow, and MC13 is green.

The online version of this article includes the following figure supplement(s) for figure 3:

**Figure supplement 1.** Characterization of T$_{FH}$ subsets using CXCR5, CXCR3, CCR6, and CD45RA expression.

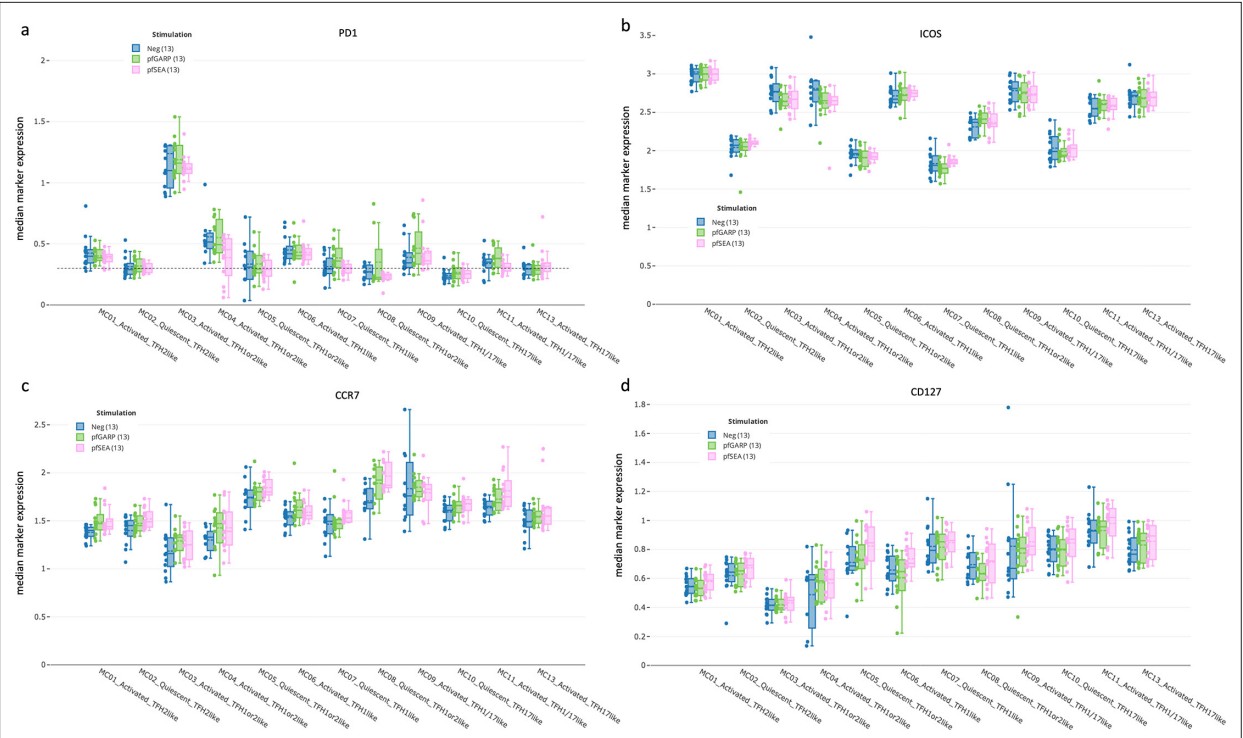

**Figure 4.** cT_FH subset activation state determined by PD1, ICOS, CCR7, and CD127. Box plots showing (**a**) PD1, (**b**) ICOS, (**c**) CCR7, and (**d**) CD127 expression across all cT_FH-like meta-clusters from children (n=13) with no stimulation (blue), and after in vitro stimulation with *Pf*GARP (green) or *Pf*SEA-1A (pink). Fifty percent of the data points are within the box limits, the solid line indicates the median, the dashed line indicates the mean, and the whiskers indicate the range of the remaining data, with outliers being outside that range.

The online version of this article includes the following figure supplement(s) for figure 4:

**Figure supplement 1.** CCR6 vs CXCR3 representative cytoplots.

**Figure supplement 2.** State of activation of cT_FH subsets from adults determined by PD1, ICOS, CCR7, and CD127.

**Figure supplement 3.** CD40L expression across stimulation and meta-clusters.

of 1000 CD3^pos CD4^pos CXCR5^pos CD25^neg cells from each sample (total of 13,000 CD3^pos CD4^pos CXCR5^pos CD25^neg cells in both children and adults). Based on CXCR3 and CCR6 expressions as well as the expression of effector/memory/activation markers (CCR7, CD127, PD1, and ICOS), cytokines (IFNγ, IL-4, and IL-21), and transcription factors (Bcl6 and cMAF), we initially identified 15 meta-clusters within the CD4^pos CXCR5^pos T cells (***Figure 3a***). First, we found different CXCR5 expression levels between meta-clusters (***Figure 3b***); CXCR5 is essential for cT_FH cells to migrate to the lymph nodes and interact with B cells. Data presented in ***Figure 3*** are from children; however, we found similar observations for adults (***Figure 3—figure supplement 1***). Because CD45RA^pos CXCR5^pos cells are likely naïve cells with transient low expression of CXCR5, yet high expression of CD45RA, we excluded three meta-clusters using these criteria (i.e. MC12, MC14, and MC15) (***Figure 3c***). Then, using the overall expression of CXCR3 and CCR6 across the cT_FH subsets (***Figure 3d and e***), we identified the remaining 12 clusters as follows: MC01 and MC02 were cT_FH2-like; MC06 and MC07 were cT_FH1-like; MC09 and MC11 were cT_FH1/17-like; MC10 and MC13 were cT_FH17-like. However, CXCR3 expression was not clearly delineated for some subsets and did not align with the conventional CCR6 vs CXCR3 cytoplot (***Figure 3f and g***, ***Figure 3—figure supplement 1***). Based on the heatmap (***Figure 3f***), MC03, MC04, MC05, and MC08 clusters appear closer to cT_FH2-like MC01 and MC02 clusters than cT_FH1-like clusters MC06 and MC07, suggesting that they might be part of the cT_FH2-like subset. But, based on their intensity of CXCR3 expression and their distribution across the CCR6 vs CXCR3 cytoplot (***Figure 3—figure supplement 1***), we defined MC03, MC04, MC05, and MC08 clusters as 'undetermined' and would require additional cytokine and transcription factor analyses to fully categorize them.

## Heterogeneity of activation/maturation markers within cT$_{FH}$ subsets

By assessing the expression of CCR7, PD1, CD127, and ICOS (*Figure 4*) and following the three-dimensional expression patterns adapted from *Schmitt et al., 2014* (*Figure 4—figure supplement 1*), we determined the activation state of each cT$_{FH}$ meta-cluster and which markers created novel subsets. Data in *Figure 4* are from children; however, similar observations were made for adults (*Figure 4—figure supplement 3*). As expected, none of the extracellular markers showed significant differences in expression patterns after a short 6 hr stimulation, thus representing the cT$_{FH}$ repertoire present within our study participants. Interestingly, the cT$_{FH}$1or2 subset (MC03) was the only meta-cluster with high expression of PD1 (*Figure 4a*) accompanied by high ICOS (*Figure 4b*), low CCR7 (*Figure 4c*), and low CD127 expression (*Figure 4d*), suggesting that MC03 was an activated/effector cT$_{FH}$1or2 subset. The MC04 subset had low CCR7 and high ICOS expression but low CD127 and intermediate PD1 expression, indicating that this cluster was a less activated/effector cT$_{FH}$1or2 cluster compared to MC03.

We also observed additional nuances in the expression pattern for other cT$_{FH}$1-like meta-clusters. The cT$_{FH}$1-like MC06 cluster had an activated/effector profile, whereas the cT$_{FH}$1-like MC07 cluster had a quiescent/effector profile. The cT$_{FH}$1or2-like MC05 and MC08 clusters were defined as quiescent/memory (overall high CCR7 expression), yet with very low expression of ICOS for the MC05 cluster.

Likewise, cT$_{FH}$2-like and cT$_{FH}$17-like meta-clusters also displayed heterogeneity. The cT$_{FH}$2-like MC01 cluster had an activated/effector profile, whereas the cT$_{FH}$2-like MC02 cluster seemed to be a quiescent/effector subset. The cT$_{FH}$1/17-like subsets MC09 and MC11 overall had an activated profile, although they had higher expression of CCR7 and CD127 compared to other subsets, suggesting an activated/memory phenotype. Finally, the cT$_{FH}$17-like MC10 cluster seemed quiescent, whereas cT$_{FH}$17-like MC13 had an activated/effector profile. Of note, because of our short stimulation time (6 hr), we were unable to find statistical differences in the CD40L expression between groups as only a few individuals responded (*Figure 4—figure supplement 3*). However, our analysis methods revealed a higher degree of previously unrecognized heterogeneity within circulating cT$_{FH}$ cells.

## Activated cT$_{FH}$1or2-, cT$_{FH}$1-, and quiescent cT$_{FH}$1or2-like subsets were more abundant in children

After having deconvoluted cT$_{FH}$ cells into 12 subsets, we next wanted to determine whether their abundance differed by age or after antigen stimulation. Using Uniform Manifold Approximation and Projection (UMAP) visualization, we found that cT$_{FH}$ dimensional reduction was contiguous as meta-clusters merged with each other; in addition, there were notable differences between adults and children (*Figure 5a*). We found that activated PD1$^{high}$ cT$_{FH}$1or2-like (MC03), activated cT$_{FH}$1-like (MC06), and quiescent ICOS$^{high}$ cT$_{FH}$1or2-like (MC08) subsets were significantly more abundant in children compared to adults regardless of the stimulation conditions (*Figure 5b, c, and d*, p<0.05). In contrast, the quiescent *Pf*SEA-1A- and *Pf*GARP-specific cT$_{FH}$2-like cluster (MC02) was significantly more abundant in adults compared to children (*Figure 5c and d*, p<0.05). Interestingly, following *Pf*GARP stimulation, the activated cT$_{FH}$1/17-like subset (MC09) became more abundant in children compared to adults (*Figure 5d*, p<0.05 with a false discovery rate [FDR] = 0.08), but no additional subsets shifted phenotype after *Pf*SEA-1A stimulation (*Figure 5c*). Of note, the activated PD1$^{low}$ cT$_{FH}$1or2-like cells (MC04) seemed more abundant in non-stimulated adults and *Pf*GARP-stimulated children, but because these observations were not present in all the participants, they did not achieve statistical significance in EdgeR.

The abundance heatmap (*Figure 5e*) reiterates the differences observed between children and adults and highlights important considerations when assessing the potential role of each cT$_{FH}$ subset in assisting with cognate antibody production. Overall, the most common cT$_{FH}$ subset in both children and adults was the quiescent cT$_{FH}$2-like cells (MC02, 24.1% and 41.2%, respectively). However, the antigen-specific differences in the cT$_{FH}$ subset abundance for children (MC09 for *Pf*GARP) and for adults (MC02 for both *Pf*SEA-1A and *Pf*GARP) suggest that children engage different cT$_{FH}$ cells as they are developing immunity. Of note, only the activated PD1$^{low}$ cT$_{FH}$1or2-like cells (MC04) were less abundant in children with low compared to high HRP2 antibody levels (*Figure 5—figure supplement 1*), suggesting that this subset may be involved in short-term antibody production. Overall, this comprehensive examination of the abundance of cT$_{FH}$ subsets demonstrates important diversity based on age and malaria-antigen specificity.

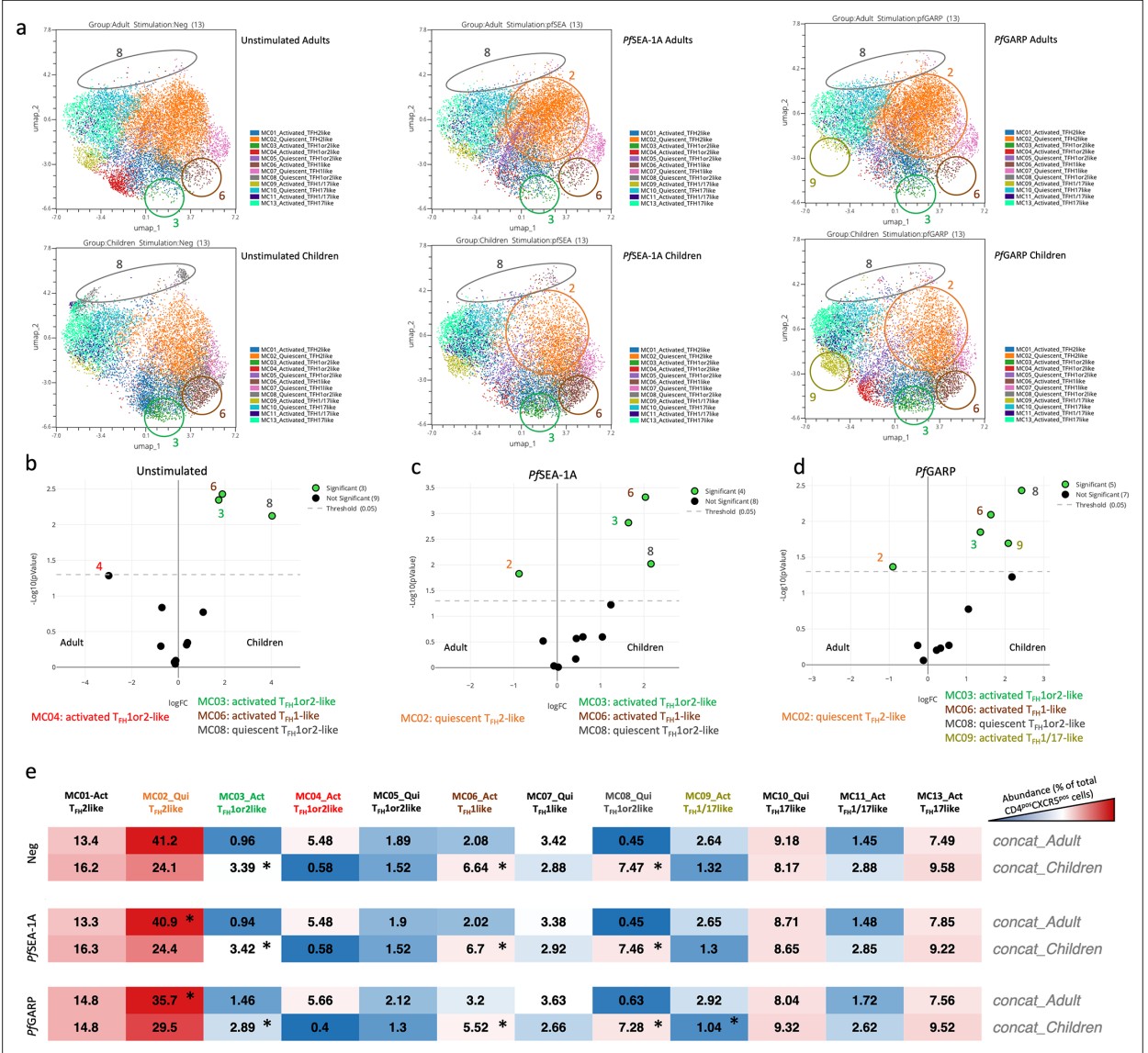

**Figure 5.** Differences in abundance of antigen-specific $cT_{FH}$ meta-clusters for adults and children. (**a**) Uniform Manifold Approximation and Projection (UMAP) plots showing the 12 different $cT_{FH}$ meta-clusters in adults (top three plots, n=13) and in children (bottom three plots, n=13) in the absence of stimulation or after in vitro stimulation with $Pf$SEA-1A or $Pf$GARP, from left to right, respectively. Colored circles highlight the meta-clusters showing differences in their abundance between adults and children for each condition. An EdgeR statistical plot was performed to assess the change in abundance of the 12 meta-clusters between adults and children after (**b**) no stimulation or stimulation with (**c**) $Pf$SEA-1A or (**d**) $Pf$GARP. EdgeR plots indicate which meta-clusters are significantly abundant between two groups by using green color dots. The Y-axis is the $-\log_{10}$(p-value), and the X-axis is the log(FC). Green dots were statistically significant ($p<0.05$). Numbers next to the dots indicate the meta-cluster. (**e**) An abundance heatmap indicates the percentage (black numbers) of each meta-cluster within the total number of $CD3^{pos}CD4^{pos}CXCR5^{pos}CD25^{neg}$ cells for adults and children (concatenated from 13 participants in each group) under the different conditions: no stimulation (Neg) or stimulation with $Pf$SEA-1A or $Pf$GARP. The color scale ranges from high expression (red) to low/no expression (blue). The star in the heatmap indicates which meta-cluster is significantly abundant in children or adults based on the EdgeR results.

The online version of this article includes the following figure supplement(s) for figure 5:

**Figure supplement 1.** Abundance of the $cT_{FH}$ meta-clusters within children with low or high levels of HRP2 antibodies.

## PfSEA-1A and PfGARP induced IL-4, Bcl6, and cMAF from a broad range of $cT_{FH}$ subsets in children

Because the children in this cohort were 7 years of age and resided in a malaria-holoendemic area, they had ample time to develop premunition. To assess antigen-specific cytokine and transcription

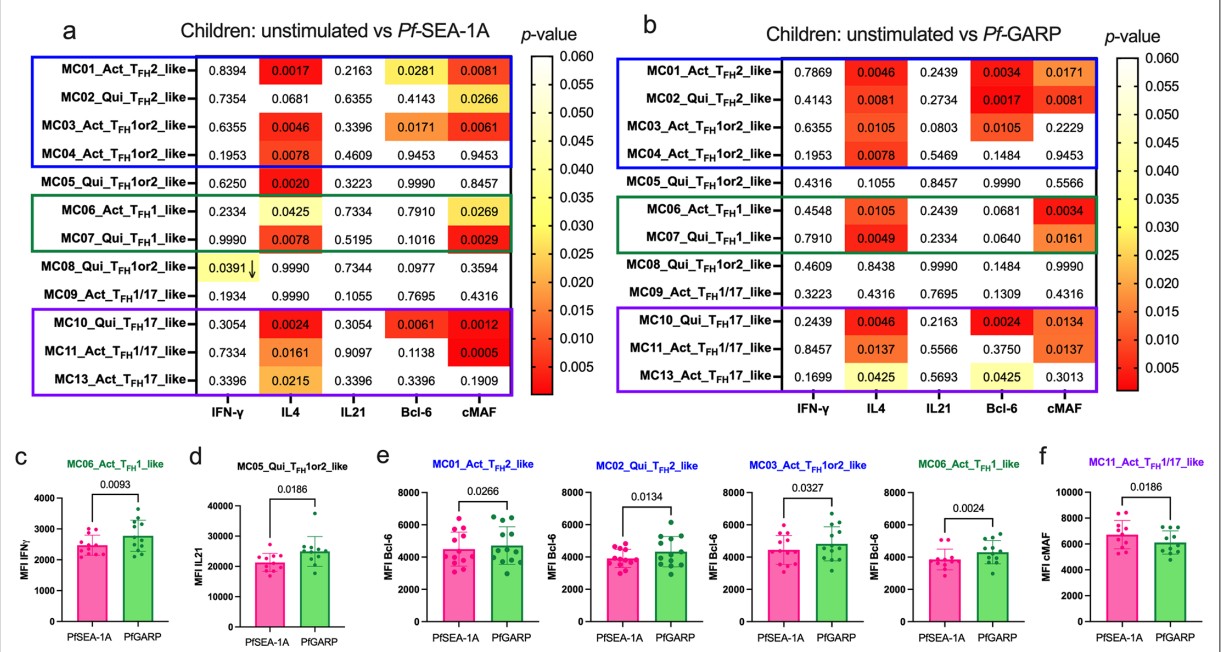

**Figure 6.** Diverse *Pf*-malaria antigen-specific expression patterns of cT$_{FH}$-defining cytokines and transcription factors for children. Heatmaps of Wilcoxon paired two-tailed t-test p-values are shown for cT$_{FH}$ meta-clusters comparing (**a**) *Pf*SEA-1A and (**b**) *Pf*-GARP vs unstimulated peripheral blood mononuclear cells (PBMCs) from children (n=13) for each cytokine (IFNγ, IL-4, and IL-21) and transcription factor (Bcl6 and cMAF). The color scale indicates the significance of the p-value: white (nonsignificant, p>0.05), yellow (0.05>p>0.02), orange (0.02>p>0.005), and red (highly significant, p<0.005). The down arrow indicates a decrease of expression from unstimulated to stimulated condition, whereas no arrow indicates an increase of expression from unstimulated to stimulated condition. The cT$_{FH}$ meta-clusters that co-expressed transcription factors were grouped as follows: Group 1 (blue line), Group 2 (green line), and Group 3 (purple line). Bar plots indicating mean with standard deviation (SD) of the median intensity fluorescence (MFI) of (**c**) IFNγ, (**d**) Bcl6, and (**e**) cMAF for the cT$_{FH}$ meta-clusters showing significant statistical differences between *Pf*SEA-1A and *Pf*GARP stimulations. The p-values from Wilcoxon paired two-tailed t-tests are indicated.

The online version of this article includes the following figure supplement(s) for figure 6:

**Figure supplement 1.** Clustered heatmap of the cytokines and transcription factors expressed from cT$_{FH}$ meta-clusters in adults and children under the different stimulation conditions.

**Figure supplement 2.** Bar plots of cytokines expressed under the different conditions of stimulation in children (n=13).

**Figure supplement 3.** Bar plots of transcription factors expressed under the different conditions of stimulation in children (n=13).

**Figure supplement 4.** Manually gated transcription factors expression upon stimulation.

**Figure supplement 5.** Manually gated IFNγ and IL21 expression upon stimulation.

**Figure supplement 6.** Manually gated IL4 and IL21 expression upon stimulation.

factor expression signatures and further characterize cT$_{FH}$ subsets, we generated clustered heatmaps of the median fluorescence intensity (MFI) of each analyte (IFNγ, IL-4, IL-21, Bcl6, and cMAF) for children (*Figure 6—figure supplement 1a*) and adults (*Figure 6—figure supplement 1b*). Next, using these MFI data, we performed Wilcoxon paired two-tailed t-tests to compare *Pf*SEA-1A and *Pf*GARP stimulation to unstimulated cells (*Figure 6a and b*, respectively). Significant differences in these expression profiles allowed us to further characterize meta-clusters into three main groups. Group 1: cT$_{FH}$2-like (activated MC01 and quiescent MC02) and activated cT$_{FH}$1or2-like (PD1$^{high}$MC03 and PD1$^{low}$MC04); Group 2: activated and quiescent cT$_{FH}$1-like (MC06 and MC07); Group 3: activated cT$_{FH}$1/17-like (MC11) and cT$_{FH}$17-like (quiescent MC10 and activated MC13). For children, *Pf*SEA-1A and *Pf*GARP induced robust IL-4 expression in 9 out of 12 cT$_{FH}$ meta-clusters (p-values≤0.0105); although the composition of which cT$_{FH}$ subsets were engaged differed slightly by antigen (quiescent cT$_{FH}$1or2-like ICOS$^{low}$ MC05 vs quiescent cT$_{FH}$2-like MC02, respectively). In contrast to IL-4, we observed no change in expression for IFNγ or IL-21 after in vitro antigen stimulation (except for quiescent cT$_{FH}$1or2-like ICOS$^{high}$ MC08, p-value=0.0391), suggesting that these cytokines are not informative to define the development of antigen-specific cT$_{FH}$ subsets in children.

We found that *Pf*SEA-1A (**Figure 6a**) and *Pf*GARP (**Figure 6b**) induced similar Bcl6 and cMAF expression profiles from some of the same cT$_{FH}$ subsets: both transcription factors were expressed by activated cT$_{FH}$2-like MC01 and quiescent cT$_{FH}$17-like MC10, but only cMAF was expressed in activated and quiescent cT$_{FH}$1-like MC06, MC07, and activated cT$_{FH}$1/17-like MC11 subsets. However, *Pf*SEA-1A induced Bcl6 and cMAF from activated PD1$^{high}$MC03, whereas *Pf*GARP only induced Bcl6. In contrast, the quiescent cT$_{FH}$2-like MC02 subset did not seem to respond to *Pf*SEA-1A (only cMAF was significant, p=0.0266, **Figure 6a**, **Figure 6—figure supplements 2 and 3**), whereas *Pf*GARP stimulation induced significantly more IL-4, Bcl6, and cMAF compared to unstimulated cells (p=0.0081, p=0.0017, and p=0.0081, respectively, **Figure 6b**, **Figure 6—figure supplements 2 and 3**). We then compared the response intensity between *Pf*SEA-1A and *Pf*GARP stimulations and found significant differences in IFNγ, IL-21, Bcl6, and cMAF expression levels (**Figure 6c, d, e, and f**, respectively). In Group 1, Bcl6 expression was significantly higher within activated and quiescent cT$_{FH}$2-like subsets (MC01 and MC02) as well as activated PD1$^{high}$ cT$_{FH}$1or2-like (MC03) cells after *Pf*GARP compared to *Pf*SEA-1A stimulation (p=0.0266, p=0.0134, and p=0.0327, respectively, **Figure 6e**). Within Group 2, IFNγ and Bcl6 were highly expressed by activated cT$_{FH}$1-like (MC06) after *Pf*GARP stimulation compared to *Pf*SEA-1A stimulation (p=0.0093 and p=0.0024, respectively, **Figure 6c and e**). Finally, within Group 3, the activated cT$_{FH}$1/17-like cells (MC11) expressed higher cMAF after *Pf*SEA-1A stimulation compared to *Pf*GARP (p=0.0186, **Figure 6f**). Interestingly, *Pf*GARP induced significantly more IL-21 within the quiescent ICOS$^{low}$ cT$_{FH}$1or2-like (MC05) meta-cluster compared to *Pf*SEA-1A stimulation (p=0.0186, **Figure 6d**). Nonsignificant differences in cytokines and transcription factors expressed by cT$_{FH}$ subsets between conditions are shown in **Figure 7—figure supplements 1 and 2**, respectively.

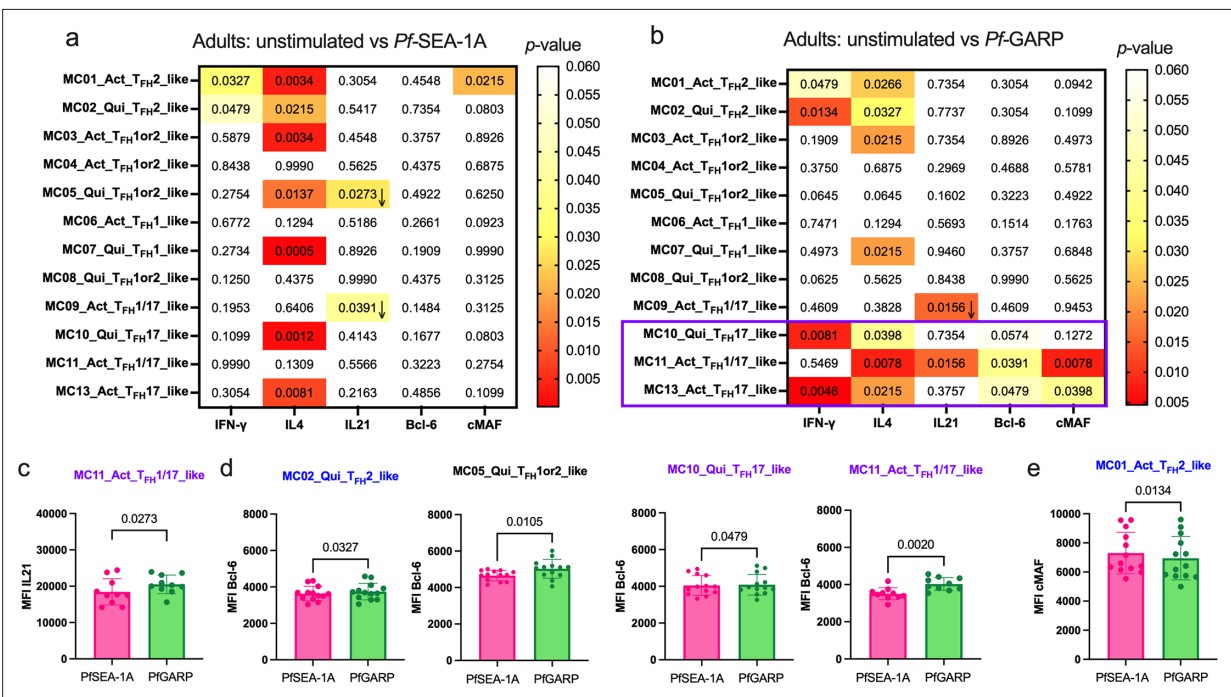

**Figure 7.** Limited *Pf*-malaria antigen-specific expression patterns of cT$_{FH}$ defining cytokines and transcription factors for adults. Heatmaps of Wilcoxon paired two-tailed t-test p-values are shown for cT$_{FH}$ meta-clusters comparing (**a**) *Pf*SEA-1A or (**b**) *Pf*-GARP vs unstimulated peripheral blood mononuclear cells (PBMCs) from adults (n=13) for each cytokine (IFNγ, IL-4, and IL-21) and transcription factors (Bcl6 and cMAF). The color scale indicates the significance of the p-value: white (nonsignificant, p>0.05), yellow (0.05>p>0.02), orange (0.02>p>0.005), and red (highly significant, p<0.005). The down arrow indicates a decrease of expression from unstimulated to stimulated condition, whereas no arrow indicates an increase of expression from unstimulated to stimulated condition. The only cT$_{FH}$ meta-clusters that expressed transcription factors were in Group 3 (purple box). Bar plots indicate the mean with standard deviation (SD) of the median fluorescence intensity (MFI) of (**c**) Bcl6 and (**d**) cMAF for the cT$_{FH}$ meta-clusters showing significant statistical differences between *Pf*SEA-1A and *Pf*GARP stimulations. The p-values from Wilcoxon paired two-tailed t-tests are indicated.

The online version of this article includes the following figure supplement(s) for figure 7:

**Figure supplement 1.** Bar plots of cytokines expressed under the different conditions of stimulation in adults (n=13).

**Figure supplement 2.** Bar plots of transcription factors expressed under the different conditions of stimulation in adults (n=13).

## PfGARP induced IL-4, Bcl6, and cMAF expression in activated cT$_{FH}$1/17- and cT$_{FH}$17-like subsets in adults

A similar heatmap was generated for adult expression profiles comparing *Pf*SEA-1A and *Pf*GARP stimulated to unstimulated cells (*Figure 7*). Here, we found that both *Pf*SEA-1A and *Pf*GARP induced significant expression of both IFNγ and IL-4 for activated (MC01) and quiescent (MC02) cT$_{FH}$2-like cells (*Figure 7a and b*, *Figure 7—figure supplement 1*). Whereas *Pf*GARP also induced IL-4 and IFNγ expression from quiescent and activated cT$_{FH}$17-like cells (MC10 and MC13), in addition to Bcl6 and cMAF for MC13 (*Figure 7b*, *Figure 7—figure supplement 2*). This observation was surprising because IFNγ expression is commonly used to categorize the cT$_{FH}$1 subset (Group 2); however, as shown earlier, quiescent and activated cT$_{FH}$17-like cells (MC10 and MC13) did not express CXCR3 (*Figure 3d and e*). In contrast, activated the cT$_{FH}$1/17-like cells (MC11) only responded to *Pf*GARP (*Figure 7b*, *Figure 7—figure supplement 1*), expressing higher levels of IL-4, IL-21, Bcl6, and cMAF. Finally, while assessing the differences between the two malaria antigens, we found that *Pf*GARP induced more Bcl6 expression than *Pf*SEA-1A within the quiescent cT$_{FH}$2-like subset (MC02, p=0.0327) and the quiescent ICOS$^{low}$ cT$_{FH}$1or2-like cells (MC05, p=0.0105), as well as within the quiescent cT$_{FH}$17-like subset (MC10, p=0.0479) and the activated cT$_{FH}$1/17-like cells (MC11, p=0.0020) (*Figure 7d*). The MC11 cells also expressed higher IL-21 levels after *Pf*GARP compared to *Pf*SEA-1A stimulation (p=0.0273, *Figure 7c*), whereas *Pf*SEA-1A induced cMAF within the activated cT$_{FH}$2-like subset (MC01, p=0.0134), but *Pf*GARP did not (*Figure 7e*). Overall, the main observation for adults is that *Pf*SEA-1A predominantly induced IL-4 from slightly more than half of the cT$_{FH}$ clusters, whereas *Pf*GARP induced a broader range of cytokines and transcription factors but within the activated cT$_{FH}$1/17-like cells (MC11) and quiescent and activated cT$_{FH}$17-like subset (MC10 and MC13) similar to children. This analysis shows clear differences in cT$_{FH}$ subset specificity by malaria antigen and cT$_{FH}$ subset engagement by age group, with the cT$_{FH}$ repertoire becoming more restricted in adults compared to children.

## The activated cT$_{FH}$1or2-like subset is more abundant in participants with high anti-PfGARP antibodies

As shown in *Figure 1b,*, a broad range of anti-*Pf*GARP IgG antibody levels were found in both children and adults. Thus, we wanted to determine whether the abundance of any of the cT$_{FH}$ subsets was associated with the level of anti-*Pf*GARP IgG antibodies. When stratifying by high vs low anti-*Pf*GARP IgG antibody levels, we found that an activated cT$_{FH}$1or2-like subset (MC04) was more abundant in

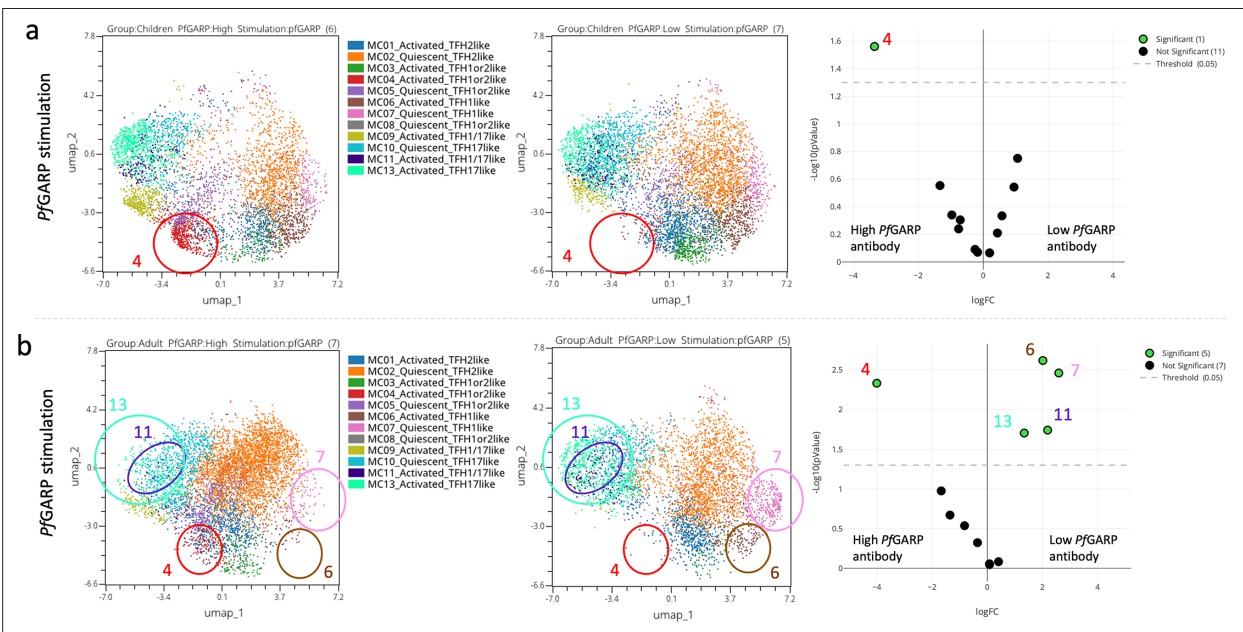

**Figure 8.** Abundance of cT$_{FH}$ meta-clusters stratified by anti-*Pf*GARP IgG antibody levels. Uniform Manifold Approximation and Projection (UMAP) and EdgeR analyses of (**a**) children (n=13) and (**b**) adults (n=13), where significant differences (p<0.05) in the abundance of the cT$_{FH}$ meta-clusters are circled on high vs low *Pf*GARP antibody level (left to right) in the UMAP plot and green dots on the volcano plot (far right).

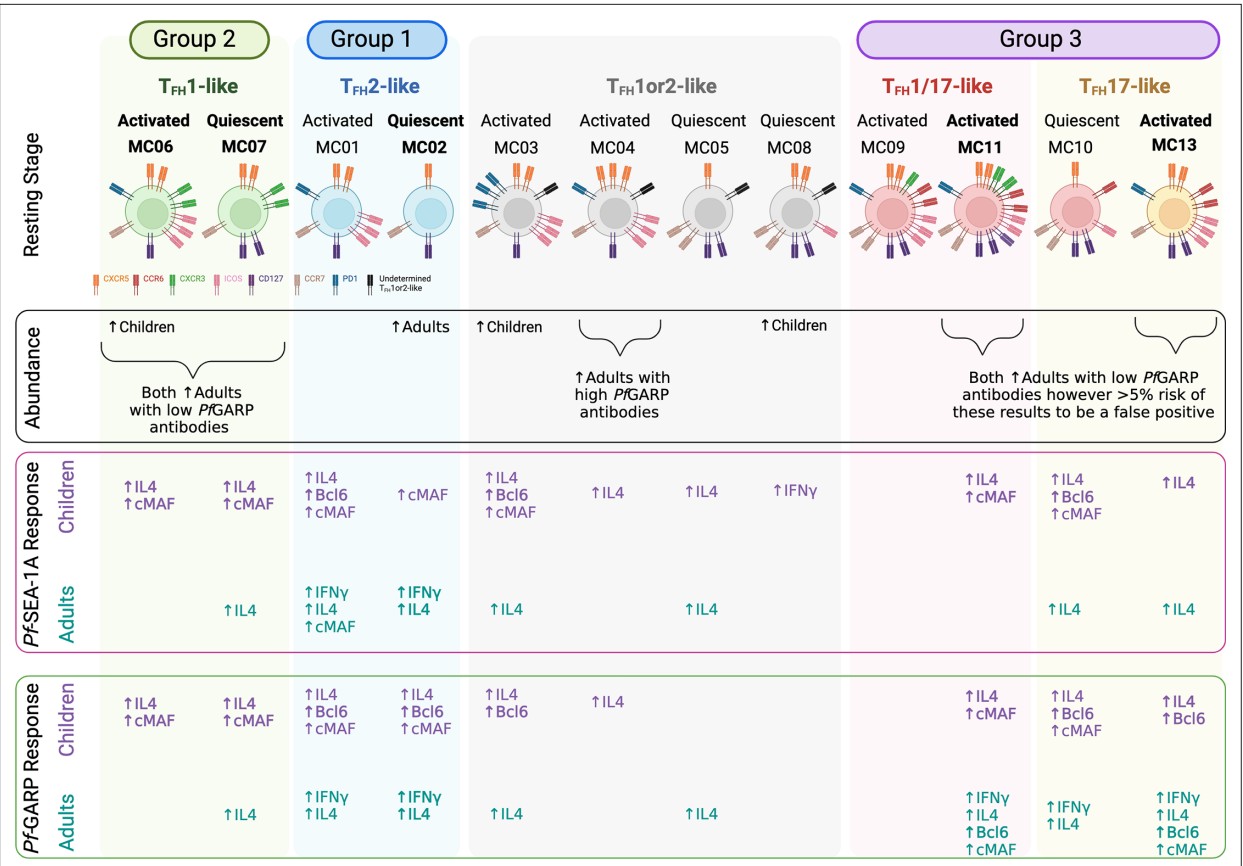

**Figure 9.** Summary. Illustration of combined findings showing remarkable differences in cT$_{FH}$ subset abundance and *Pf*-malaria antigen-specific cytokine and transcription factor responses between children and adults residing in a malaria-holoendemic region of Kenya. Figure created with BioRender.

participants with high levels of anti-*Pf*GARP IgG for both children (*Figure 8a*) and adults (*Figure 8b*) after *Pf*GARP stimulation compared to participants with low levels or an absence of anti-*Pf*GARP IgG. However, even though the p-values were significant for both children and adults (p=0.02 and p=0.004, respectively), the FDR was less than 0.05 only for the adults (FDR = 0.018). This suggests that this particular subset might be important for the generation of anti-*Pf*GARP antibodies. Interestingly, activated cT$_{FH}$1-like (MC06) and quiescent cT$_{FH}$1-like (MC07) cells were more abundant for adults with low or no anti-*Pf*GARP IgG antibodies compared to those with high levels (p=0.0024 with FDR = 0.0185 and p=0.0034 with FDR = 0.0185, respectively), consistent with previous observations describing T$_{FH}$1 subsets as inefficient help for antibody production (*Obeng-Adjei et al., 2015*).

## Discussion

The overall aim of this study was to define cT$_{FH}$ subsets using unbiased clustering analysis and assess their malaria antigen-specific (*Pf*SEA-1A and *Pf*GARP) (*Raj et al., 2014*; *Raj et al., 2020*; *Duffy and Patrick Gorres, 2020*) profiles for adults and children residing in a malaria-holoendemic area of Western Kenya. Contrary to the previous publications (*Obeng-Adjei et al., 2015*; *Oyong et al., 2022*), our study found that children not only respond via their cT$_{FH}$1-like subsets but also engage a broader spectrum of cT$_{FH}$ subsets. In fact, cytokine and transcription factor profiles for children involved cT$_{FH}$1-, cT$_{FH}$2-, cT$_{FH}$17-, and cT$_{FH}$1/17-like subsets (summarized in *Figure 9*), whereas in adults, the dominant antigen-specific memory response was from cT$_{FH}$17- and cT$_{FH}$1/17-like subsets but only for *Pf*GARP. Thus, revealing a potential difference in engaging cT$_{FH}$ help between the two malaria vaccine candidates evaluated in this study.

Several key points can be made from our results. First, it may be too soon in the field of cT$_{FH}$ biology to select only one or two types of cT$_{FH}$ subset(s) to measure when evaluating malaria vaccine

candidates, especially if we miss essential, albeit transient, correlates of protection for children which may differ in adults. Longitudinal studies with clinical outcomes are needed to determine the impact of the changing dynamics of cT$_{FH}$ subset adaptations after repeated malaria infections that lead to premunition and parasite clearance. A Ugandan study by *Chan et al., 2022*, demonstrated an age-associated change in cT$_{FH}$ subsets independent of malaria and, thus, supports the importance of accounting for age when evaluating antigen-specific cT$_{FH}$ profiles. Second, our study reveals that adults have a robust cT$_{FH}$17- and cT$_{FH}$1/17-like subset response to *Pf*GARP but not to *Pf*SEA-1A. In addition to demonstrating differences between malaria antigens, this finding supports the premise that cT$_{FH}$17 cells are important for maintaining immunological memory (*Gao et al., 2023*), which has important implications for evaluating malaria vaccine efficacy. Third, we found a correlation between cT$_{FH}$1or2-like (MC04) and high *Pf*GARP antibodies for both children and adults, indicating that a cT$_{FH}$ subset accompanied by high antibody levels could serve as a potential biomarker of protection. More studies are needed to explore this association; however, we postulate that a coalition of cT$_{FH}$ subsets might be engaged to develop long-lived antibody responses, and therefore, categorizing subsets as efficient or inefficient might be context-dependent (*Obeng-Adjei et al., 2015*).

As the field of computational immunology and unbiased clustering analyses evolves, it presents an ongoing challenge to meaningfully define cT$_{FH}$ subsets. Here, we intentionally chose to use cT$_{FH}$1or2-like nomenclature for meta-clusters MC03, MC04, MC05, and MC08 (*Figure 9*) because of their low (but not null) CXCR3 and robust IL-4 expression along with Bcl6 and cMAF. CXCR3 expression leans toward a cT$_{FH}$1-like polarization, whereas IL-4 expression indicates a cT$_{FH}$2-like profile when we follow the most commonly used cT$_{FH}$ classification (*Schmitt et al., 2014*; *Seth and Craft, 2019*; *Olatunde et al., 2021*; *Bentebibel et al., 2015*). This difference is crucial as cT$_{FH}$2 cells are described as good promoters for functional GC antibodies (*Chan et al., 2020*; *Oyong et al., 2022*), whereas cT$_{FH}$1 cells are not (*Obeng-Adjei et al., 2015*; *Oyong et al., 2022*; *Hansen et al., 2017*). Therefore, their definitive classification will require a more in-depth investigation. Although we used malaria antigen-specific stimulation, and not a *Pf*-lysate or iRBCs as previously described (*Obeng-Adjei et al., 2015*; *Oyong et al., 2022*; *Chan et al., 2022*), adults had significantly more of the quiescent cT$_{FH}$2-like subset (MC02) compared to children who instead had significantly more of the activated cT$_{FH}$1-like subset (MC06), the latter being consistent with the previous publications (*Obeng-Adjei et al., 2015*; *Oyong et al., 2022*), and more PD1$^{high}$ activated and ICOS$^{high}$ quiescent cT$_{FH}$1or2-like subsets (MC03 and MC08, respectively). However, even when classifying the cT$_{FH}$ meta-clusters as cT$_{FH}$1, cT$_{FH}$2, cT$_{FH}$17, or cT$_{FH}$1/17, the most abundant cT$_{FH}$ subset was cT$_{FH}$2 cells for adults (more than 50%), contradicting a previous study showing only ~20% of cT$_{FH}$2 cells for the same age range (*Chan et al., 2022*). As suggested by Gowthaman's T$_{FH}$ model (*Gowthaman et al., 2019*), T$_{FH}$1 cells are involved in the development of neutralization antibody responses to viruses and bacteria, whereas parasites, such as helminths, lead to a T$_{FH}$2 response. These observations reinforce the need to assess T$_{FH}$ subset profiles stratified by exposure to potentially immune-modulating co-infections.

Transcription factors cMAF and Bcl6 play essential roles in T$_{FH}$ development and function (*Nurieva et al., 2009*; *Imbratta et al., 2020*; *Liu et al., 2012*). cMAF induces the expression of various molecules, such as ICOS, PD1, CXCR5, IL-4, and IL-21, which are all essential for T$_{FH}$ function (*Imbratta et al., 2020*). It was, therefore, not surprising to find that cMAF significantly increased after antigen stimulation for most of the cT$_{FH}$ subsets in children. However, again, this expression profile was only observed in the activated cT$_{FH}$1/17-like subset (MC11) after *Pf*GARP stimulation in cells from adults, supporting a role for T$_{FH}$17 cells in the maintenance of immunological memory (*Gao et al., 2023*). Importantly, cMAF cooperates with Bcl6 in T$_{FH}$ development and function and is essential to establish an efficient GC response (*Nurieva et al., 2009*; *Yu et al., 2009*). Bcl6 is also described as being expressed by mature T$_{FH}$ cells, inducing the expression of CXCR5 and PD1, but it can also regulate IFNγ and IL-17 production (*Crotty, 2014*; *Nurieva et al., 2009*; *Yu et al., 2009*; *Johnston et al., 2009*). In our study, *Pf*GARP induced a highly significant increase of Bcl6 in cT$_{FH}$2-like and cT$_{FH}$17-like subsets in children, suggesting that *Pf*GARP may be a better candidate to trigger an efficient humoral response in children compared to *Pf*SEA-1A. Of note, Blimp1, a Bcl6 antagonist (*Johnston et al., 2009*), was absent from our flow panel and would be of interest to assess in future studies.

There are several limitations of human immune profiling studies. Similar to other such studies, we used peripheral blood and, thus, were only able to provide a snapshot of the cT$_{FH}$ cells. Study participants did not have blood-stage malaria infections at the time of blood collection, and children were

7 years of age; thus, our profiles were by design meant to reflect $cT_{FH}$ memory recall responses and demonstrate differences between children and adults. As this was not a birth cohort study design, the number of cumulative malaria infections could have confounded the association between malaria and $cT_{FH}$ subsets, causing a certain $cT_{FH}$ meta-cluster to arise. Most human immunology studies are unable to assess tissue-resident $T_{FH}$ or $T_{FH}$ in the lymph nodes. Therefore, we can only speculate on the composition of $T_{FH}$ cells observed in children that were perhaps short-lived helper cells and not maintained as T cell memory cells or may have trafficked from the blood into tissues over time (*Potter et al., 2021*). Using an in vivo nonhuman primate model, Potter et al. showed an entry rate of lymphocyte subsets into peripheral lymph nodes per hour of 1.54% and 2.17% for CD4$^{pos}$ central memory (CD45$^{neg}$CCR7$^{pos}$) and effector memory (CD45$^{neg}$CCR7$^{neg}$) T cells, respectively (*Potter et al., 2021*), both of which may include central and effector memory $cT_{FH}$ cells. With this in mind, $cT_{FH}$ cells seem to be at a crossroads with multiple possible fates. The first one being the recirculation of $cT_{FH}$ cells from lymph node to lymph node, whereby a peripheral blood sampling captures migratory $cT_{FH}$ subsets, which could explain the $cT_{FH}$ subsets we observed still expressing cMAF and Bcl6. A decade ago, it was hypothesized that once a $T_{FH}$ cell leaves the lymph node, it can become a PD1$^{low}$ memory $cT_{FH}$ cell with the possibility of returning to a GC $T_{FH}$ stage after a secondary recall response (*Crotty, 2014*). Other possible fates include becoming a PD1$^{neg}$ memory $cT_{FH}$ with the same path after recall or progressing to a non-$T_{FH}$ cell (*Crotty, 2014*). More recently, a few studies showed circulating CXCR5$^{neg}$CD4$^{pos}$ cells with $T_{FH}$ functions, such as production of IL-21 and the capability to help B cells in systemic lupus erythematosus and HIV-infected individuals (*Bocharnikov et al., 2019*; *Del Alcazar et al., 2019*), and importantly researchers mapped these circulating CXCR5$^{neg}$CD4$^{pos}$ cells to an original lymph node CXCR5$^{pos}$ $T_{FH}$ subset (*Del Alcazar et al., 2019*), suggesting another possible outcome for $T_{FH}$ cells outside secondary lymphoid organs. To date, no clear destiny has been established for $cT_{FH}$ cells in humans, thus highlighting the importance of including a complete panel of $cT_{FH}$ subsets to continue to improve our understanding of their respective roles against different pathogens and eliciting long-lived vaccine-induced antibody responses.

Other limitations of this study include not measuring other cytokines (i.e. IL-5, IL-13, and IL-17) and transcription factors (i.e. T-bet, BATF, GATA3, and RORγt) that have been used in other studies to fully characterize $cT_{FH}$ subsets (*Crotty, 2014*; *Gowthaman et al., 2019*). Although the heterogeneity in the response of CD40L and IFNγ suggests that our tested malaria antigens did not induce significant differences in the expression of these markers in all our participants, our panel did not include other activated induced markers, such as OX40, 4-1BB, and CD69. However, even with our small sample size, we demonstrated significant age-associated differences in malaria antigen-specific responses from different $cT_{FH}$ subsets. To minimize false-positive results that can arise when using algorithms for computational analyses, we ran the statistical tests in triplicate and the clustering algorithm in duplicate to validate our findings.

In summary, our study provides additional justification for the resources needed to conduct cellular immunological studies of $cT_{FH}$ cell signatures and provides insight into which types of $cT_{FH}$ subsets during a person's lifespan assist B cells in the GC to produce long-lived plasmocytes and functional antibodies (*Moormann et al., 2019*) against malaria. This is particularly important when selecting immune correlates of protection that could be used to predict the efficacy of the next generation of *Pf*-malaria vaccine candidates within various study populations.

## Methods

**Key resources table**

| Reagent type (species) or resource | Designation | Source or reference | Identifiers | Additional information |
|---|---|---|---|---|
| Biological sample *Homo sapiens* | PBMCs, Plasma, cell pellet | From our studied cohort | N/A | Used 1 million PBMCs per condition |
| Antibody | Co-stimulatory antibodies CD28/CD49d | BD | Cat# 347690 | Fast Immune 5 µl per test |
| Antibody | CCR6BV421, Mouse, clone 561 | BioLegend | Cat#343610, RRID:AB_2561356 | 1 µl per million PBMCs |

*Continued on next page*

*Continued*

| Reagent type (species) or resource | Designation | Source or reference | Identifiers | Additional information |
|---|---|---|---|---|
| Antibody | CD14-Pacific Blue, Mouse, clone HCD14 | BioLegend | Cat#325616, RRID:AB_830689 | 1 µl per million PBMCs |
| Antibody | CD19-Pacific Blue, Mouse, clone HIB19 | BioLegend | Cat#302232, RRID:AB_2073118 | 1 µl per million PBMCs |
| Antibody | CCR7-BV480, Rat, Clone 3D12 | BD | Cat#566099, RRID:AB_2739502 | 4 µl per million PBMCs |
| Antibody | IFNγ-BV510, Mouse, Clone 4SB3 | BioLegend | Cat#502544, RRID:AB_2563883 | 1 µl per million PBMCs |
| Antibody | CD127-BV570, Mouse, Clone A019D5 | BioLegend | Cat#351307, RRID:AB_10900064 | 1 µl per million PBMCs |
| Antibody | CD45RABV605, Mouse, Clone HI100 | BioLegend | Cat#304134, RRID:AB_2563814 | 0.2 µl per million PBMCs |
| Antibody | PD1-BV650, Mouse, Clone MIH4 | BD | Cat#564324, RRID:AB_2738746 | 4 µl per million PBMCs |
| Antibody | CXCR3-BV711, Mouse, Clone G025H7 | BioLegend | Cat#353732, RRID:AB_2563533 | 4 µl per million PBMCs |
| Antibody | CD25-BV750, Mouse, Clone M-A251 | BD | Cat#747158, RRID:AB_2871896 | 2 µl per million PBMCs |
| Antibody | CXCR5-BV785, Mouse, Clone J252D4 | BioLegend | Cat#356936, RRID:AB_2629528 | 2 µl per million PBMCs |
| Antibody | Bcl6-AF488, Mouse | BD | Cat#561524, RRID:AB_10716202 | 5 µl per million PBMCs |
| Antibody | CD3-Spark Blue 550, Mouse, clone SK7 | BioLegend | Cat#344852, RRID:AB_2819985 | 0.2 µl per million PBMCs |
| Antibody | CD8-PerCP-Cy5.5, Mouse, clone SK1 | BioLegend | Cat#344710, RRID:AB_2044010 | 0.02 µl per million PBMCs |
| Antibody | IL-21-PE, Mouse, clone 3A3-N2 | BioLegend | Cat#513004, RRID:AB_2249025 | 10 µl per million PBMCs |
| Antibody | IL-4-PE-Dazzle, Rat, clone MP4-25D2 | BioLegend | Cat#500832, RRID:AB_2564036 | 4 µl per million PBMCs |
| Antibody | CD4-PE-Cy5, Mouse, clone RPA-T4 | BioLegend | Cat#300510, RRID:AB_314078 | 0.04 µl per million PBMCs |
| Antibody | ICOS-PE-Cy7, Armenian Hamster, clone C398.4A | BioLegend | Cat#313520, RRID:AB_10643411 | 1 µl per million PBMCs |
| Antibody | cMAF-eFluor 660, Mouse, clone sym0F1 | Thermo Fisher | Cat#50985582, RRID:AB_2574388 | 2 µl per million PBMCs |
| Antibody | CD40L-AF700, Mouse, clone 24–31 | BioLegend | Cat#310846, RRID:AB_2750053 | 2 µl per million PBMCs |
| Antibody | Biotinylated anti-human IgG | BD | Cat#555785 | Diluted 1:1000 |
| Peptide, recombinant protein | *Plasmodium falciparum* malaria antigens | Kurtis' lab Brown University | *Pf*-GARP and *Pf*-SEA-1A | 5 µg/ml (PfSEA-1A) 10 µg/ml (PfGARP) |
| Peptide, recombinant protein | *Plasmodium falciparum* malaria antigens | Walter Reed Army Institute of Research | AMA1, MSP1, CelTos, HRPII | 100 µg of each |
| Commercial assay or kit | Transcription factor buffer set Fix/Perm | BD | Cat#562574 | Followed manufacturer's instructions |
| Chemical compound, drug | Bovine Serum Albumin | Sigma | Cat#A3294 | Solution of 1 mg/ml |
| Chemical compound, drug | Staphylococcal enterotoxin B | EMD Millipore | Cat# 324798 | Used at 1 µg/ml final |

*Continued on next page*

*Continued*

| Reagent type (species) or resource | Designation | Source or reference | Identifiers | Additional information |
|---|---|---|---|---|
| Chemical compound, drug | GolgiPLUG Brefaldin A | BD | Cat# 555029 | Used at 0.1 µg/ml |
| Chemical compound, drug | GolgiSTOP Monensin | BD | Cat# 554724 | Used at 0.7 µg/ml |
| Chemical compound, drug | Streptavidin-PE Detection | BD | Cat#554061 | Diluted 1:1000 |
| Software, algorithm | SpectroFlow | Cytek | N/A | |
| Software, algorithm | OMIQ platform | OMIQ | N/A | |
| Software, algorithm | GraphPad Prism | GraphPad | N/A | version 7.0 |
| Other | Zombie NIR | BioLegend | Cat# 423106 | Live/Dead staining 1:1000 |
| Other | Ultra-Compensation beads Plus | Thermo Fisher eBiosciences | Cat#01333342 | 1 drop per compensation control |

## Study populations and ethical approvals

Adults and children were recruited from Kisumu County, Kenya, which is holoendemic for *Pf*-malaria. Written informed consent was obtained from each adult participant and every child's guardian. An abbreviated medical history, physical examination, and blood film were used to ascertain health and malaria infection status at the time of blood sample collection. Participants were also lifelong residents of the study area with the assumption that they naturally acquired immunity to malaria. This study was conducted before the implementation of any malaria vaccines. Participants were eligible if they were healthy and not experiencing any symptoms of malaria at the time venous blood was collected. For this cross-sectional immunology study, we selected fourteen 7-year-old children from a larger age-structured prospective cohort study (enrollment age range 3–7 years) and fifteen Kenyan adults. We selected 7-year-olds because of the age-dependent shift in major cT$_{FH}$ subsets occurring after 6 years of age (*Chan et al., 2022*), and as a comparable age published by other studies (*Obeng-Adjei et al., 2015*), to maximize our ability to measure antigen-specific differences in T$_{FH}$ subsets.

Ethical approvals were obtained from the Scientific and Ethics Review Unit (SERU) at the Kenya Medical Research Institute (KEMRI) reference number 3542, and the Institutional Review Board at the University of Massachusetts Chan Medical School, Worcester, MA, USA, IRB number H00014522. Brown University, Providence, RI, USA, signed a reliance agreement with KEMRI.

## Plasma and PBMC isolation

Venous blood was collected in sodium heparin BD Vacutainers and processed within 2 hr at the Center for Global Health Research, KEMRI, Kisumu. ALCs were determined from whole blood using the BC-3000 Plus Auto Hematology Analyzer, 19 parameters (Shenzhen Mindray Bio-Medical Electronics Co). After 10 min at 1000×*g* spin, plasma was removed and stored at –20°C, and an equivalent volume of 1× PBS was added to the cell pellet. PBMCs were then isolated using Ficoll-Hypaque density gradient centrifugation on SepMate (StemCell). PBMCs were frozen at 5×10$^6$ cells/ml in a freezing medium (90% heat-inactivated and filter-sterilized fetal bovine serum [FBS] and 10% dimethyl sulfoxide [Sigma]) and chilled overnight in Mr. Frosty containers at –80°C before being transferred to liquid nitrogen. For transport to the USA, an MVE vapor shipper (MVE Biological Solutions) was used to maintain the cold chain.

## In vitro stimulation assay

PBMCs were thawed in 37°C filtered-complete media (10% FBS, 2 mM L-glutamine, 10 mM HEPES, 1× penicillin/streptomycin) and spun twice before resting overnight in a 37°C, 5% CO$_2$ incubator. PBMCs were counted using Trypan Blue (0.4%) and a hemocytometer, and the cell survival was calculated. Our samples showed a median of 94.6% live cells (25% percentile of 92%; 75% percentile of 97%). Using a P96 U-bottom plate, 1×10$^6$ PBMCs per well were placed in culture with one of the following stimulation conditions: *Pf*SEA-1A (*Raj et al., 2014*) (5 µg/ml) or *Pf*GARP (*Raj et al., 2020*) (10 µg/ml) both produced in the Kurtis lab (Brown University); SEB (1 µg/ml; EMD Millipore) was used as a positive control; sterile water (10 µl, the same volume used to reconstitute *Pf*SEA and *Pf*GARP) was used as a negative control. A pool of anti-CD28/anti-CD49d (BD Fast-Immune Co-Stim following

the manufacturer's instructions), GolgiSTOP (0.7 µg/ml), and GolgiPLUG (0.1 µg/ml) (BD Biosciences) was added to each well before incubating cells at 37°C for 6 hr.

## Cell staining and flow cytometry

A multiparameter spectral flow cytometry panel was used to characterize $cT_{FH}$ cell subsets: CCR6-BV421 (RRID:AB_2561356), CD14-Pacific Blue (RRID:AB_830689), CD19-Pacific Blue (RRID:AB_2073118), CCR7-BV480 (RRID:AB_2739502), IFNγ-BV510 (RRID:AB_2563883), CD127-BV570 (RRID:AB_10900064), CD45RA-BV605 (RRID:AB_2563814), PD1-BV650 (RRID:AB_2738746), CXCR3-BV711 (RRID:AB_2563533), CD25-BV750 (RRID:AB_2871896), CXCR5-BV785 (RRID:AB_2629528), Bcl6-AF488 (RRID:AB_10716202), CD3-Spark Blue 550 (RRID:AB_2819985), CD8-PerCP-Cy5.5 (RRID:AB_2044010), IL-21-PE (RRID:AB_2249025), IL-4-PE-Dazzle (RRID:AB_2564036), CD4-PE-Cy5 (RRID:AB_314078), ICOS-PE-Cy7 (RRID:AB_10643411), cMAF-eFluor 660 (RRID:AB_2574388), CD40L-AF700 (RRID:AB_2750053), and Zombie NIR (BioLegend cat# 423106) for Live/Dead staining. Cells were fixed and permeabilized for 45 min using the transcription factor buffer set (BD Pharmingen) followed by a wash with the perm-wash buffer. Intracellular staining was performed at 4°C for 45 more minutes followed by two washes using the kit's perm-wash buffer. Data was acquired on a Cytek Aurora with 4 lasers (UMass Chan Flow Core Facility) using SpectroFlo software (Cytek) and compensation for unmixing and fluorescence-minus-one controls. Quality control of the data was performed using SpectroFlo, and the multiparameter analysis was performed with OMIQ data analysis software (https://www.omiq.ai/). Thus, we assessed the expression of markers commonly used to define the following different $cT_{FH}$ ($CD4^{pos}CD25^{neg}CXCR5^{pos}$) subsets: $cT_{FH}$1-like ($CCR6^{neg}CXCR3^{pos}$), $cT_{FH}$2-like ($CCR6^{neg}CXCR3^{neg}$), and $cT_{FH}$17-like ($CCR6^{pos}CXCR3^{neg}$) (*Schmitt et al., 2014*), as well as quiescent/central memory $cT_{FH}$ ($CCR7^{high}PD-1^{neg}ICOS^{neg}$) or activated/effector memory $cT_{FH}$ cells ($CCR7^{low}PD-1^{pos}ICOS^{pos}$) (*Schmitt et al., 2014*; *Gong et al., 2019*). Representative cytoplots can be found in *Figure 2—figure supplement 1*.

## Multiplex suspension bead-based serology assay

To measure plasma IgG antibody levels to *Pf*SEA-1A (*Raj et al., 2014*) and *Pf*GARP (*Raj et al., 2020*), we used a Luminex bead-based suspension assay as previously published (*Cham et al., 2009*; *Forconi et al., 2018*). In addition, previous *Pf* exposure was determined using recombinant proteins to blood-stage malaria antigens: AMA1, MSP1, HRP2, CelTos, and CSP (gifts from Sheetji Dutta, Evelina Angov, and Elke Bergmann from the Walter Reed Army Institute of Research). Briefly, 100 µg of each antigen or BSA (Sigma), as a background control, was coupled to ~12 × 10^6 nonmagnetic microspheres (Bio-Rad carboxylated beads) and then incubated with study participant plasma (spun down 10,000×*g* for 10 min and diluted at 1:100 in the assay dilution buffer) for 2 hr, followed by incubation with biotinylated anti-human IgG (BD #555785) diluted 1:1000 for 1 hr and streptavidin (BD #554061) diluted 1:1000 for 1 hr following the manufacturer's instructions. The MFI of each conjugated bead (minimum of 50 beads per antigen) was quantified on a FlexMap3D Luminex multianalyte analyzer (Xponent software). Results are reported as antigen-specific MFI after subtracting the BSA value for each individual since background levels can vary between individuals.

## OMIQ analysis

The fcs files were uploaded into the OMIQ platform after passing quality control under SpectroFlo (Cytek) where compensation was re-checked. In OMIQ, we arcsinh-transformed the scale to allow downstream analysis and then gated on singlet live lymphocytes and subsampled the data to yield 87,712 live lymphocytes per sample. Using only lineage markers CD3, CD4, CD8, CD14, CD19, CXCR5, and CD25, FlowSOM consensus meta-clustering was run on 100 clusters based on the 87,712 live lymphocytes per sample with a comma-separated k-value of 75 and Euclidean distance metric. Using these 75 meta-clusters, we defined subsets of cells based on lineage markers, such as $CD3^{pos}CD8^{pos}$, $CD3^{neg}CD14^{pos}CD19^{pos}$, and $CD3^{pos}CD4^{pos}$, and then distinguished $CD4^{pos}CXCR5^{pos}CD25^{neg}$ ($cT_{FH}$) and $CD4^{pos}CXCR5^{neg}CD25^{pos}$ (T regulatory [$T_{reg}$] or T-follicular regulatory [$T_{FR}$]) subsets. CD25 was used to exclude $T_{reg}$ and $T_{FR}$ cells which share numerous markers with $cT_{FH}$ cells (*Wing et al., 2018*; *Sage et al., 2014*; *Zhao et al., 2020*). EmbedSOM dimensional reduction was used to visualize the different groups of cells, and EdgeR analysis was run to assess the significance of their differences. A clustering heatmap was used to visualize cytokine expression and transcription factor profiles for each subset.

Focusing on the CD4$^{pos}$CXCR5$^{pos}$CD25$^{neg}$ T$_{FH}$ cells, we ran another FlowSOM analysis based on the 1000 CXCR5$^{pos}$ cells per sample (two samples from the adult group and one sample from the children group were excluded from the analysis as they had less than 1000 CXCR5$^{pos}$ cells), using extracellular markers CXCR5, CXCR3, CCR6, ICOS, CCR7, CD45RA, CD127, CD40L, and PD1 enabled the identification of 15 meta-clusters. If the fcs file had more than 1000 CXCR5$^{pos}$ cells, the down-sampling was done randomly by the OMIQ platform algorithm to select only 1000 CXCR5$^{pos}$ cells within this specific fcs file. From there, we performed UMAP dimensional reduction, heatmaps, and EdgeR analyses; the latter allowed statistical analysis of the cT$_{FH}$ abundances. To demonstrate the reproducibility of these results, statistical analysis algorithms were run at least three times downstream of the same clustering algorithm and downstream of repeated clustering algorithms. To assess statistical differences in cytokines and transcription factor expression, we exported the statistics dataset from OMIQ containing MFI values from each marker (IFNγ, IL-4, IL-21, Bcl6, and cMAF) per cluster and for each sample and stimulation condition. To assess cytokines and transcription factors without bias, we chose to use the total MFI expression per meta-cluster with the assumption that cells with increased production of the desired analyte trigger an increase in the overall meta-cluster MFI compared to unstimulated cells, and if there is no production of the desired analyte, the overall MFI will not differ. However, the percentage of positive IFNγ, IL-4, IL-21, Bcl6, or cMAF using manual gating can be found in *Figure 6—figure supplements 4–6* along with the overlay of the gated positive cells on the CD4$^{pos}$CXCR5$^{pos}$CD25$^{neg}$ UMAP and the cytoplots of the gated positive cells for each meta-cluster (*Figure 6—figure supplements 4–6*).

## Statistical analysis

For this cross-sectional immunology study, we selected both male and female study participants (sex defined at birth). There were fourteen 7-year-old children and fifteen adults. Using GraphPad Prism software (version 7.0), age, sex, ALC, and serological data were compared between adults and children. Because the number of participants within each group was too low to verify the normality of the underlying distributions (adults n=15 and children n=14), we chose to use non-parametric tests, including the Mann-Whitney U test (for unpaired analysis) and Wilcoxon signed-rank test (for paired analysis). When data passed the normality test (D'Agostino and Pearson test), we used Welch's parametric test. All tests were two-tailed with a p-value<0.05 for significance. Because of the exploratory nature of the analysis, we did not use any adjustment of the p-value for multiple comparisons. The tests used are indicated in the legend of each figure. Results were expressed as the mean with SD and exact p-values for dot plots.

To perform a statistical analysis of the cytokine and transcription factor expression from each cT$_{FH}$ subset, the exported data file from OMIQ was integrated into GraphPad Prism, and a non-parametric Wilcoxon paired two-tailed t-test analysis was done (n=13 in each group). As this analysis generated numerous bar plots (all included in the *Figure 6—figure supplements 4–6*), to better visualize the cytokines and transcription factor patterns, the p-values obtained from each analysis are presented using a non-clustering heatmap.

## Acknowledgements

The authors would like to thank the children and their families for participating in this study. We thank the field and lab teams for their work collecting data and processing blood samples. We also thank Dr. Melanie Trombly from UMass Chan for proofreading our manuscript. This manuscript was approved for publication by KEMRI. This study was supported by NIH R01 AI127699 (Kurtis).

## Additional information

### Competing interests

Jonathan D Kurtis: Principal investigator on 1R01AI127699-01A1, which supported this study; holds several patents related to the use of PfSEA-1 and PfGARP as vaccine candidates for P. falciparum and has consulted for and is an equity holder in Ocean Biomedical. The other authors declare that no competing interests exist.

## Funding

| Funder | Grant reference number | Author |
|--------|------------------------|--------|
| National Institutes of Health | R01AI127699-01A1 | Jonathan D Kurtis |

The funders had no role in study design, data collection and interpretation, or the decision to submit the work for publication.

## Author contributions

Catherine Suzanne Forconi, Formal analysis, Methodology, Writing – original draft, Writing – review and editing; Christina Nixon, Conceptualization, Investigation, Methodology; Hannah W Wu, Data curation, Supervision, Methodology, Project administration; Boaz Odwar, Sunthorn Pond-Tor, Data curation, Project administration; John M Ong'echa, Conceptualization, Resources, Supervision, Validation, Investigation, Project administration; Jonathan D Kurtis, Conceptualization, Supervision, Funding acquisition, Validation, Investigation, Writing – review and editing; Ann M Moormann, Conceptualization, Supervision, Validation, Investigation, Methodology, Writing – original draft, Writing – review and editing

## Author ORCIDs

Catherine Suzanne Forconi ⬡ https://orcid.org/0000-0001-7358-5546
John M Ong'echa ⬡ https://orcid.org/0000-0003-3928-6774
Jonathan D Kurtis ⬡ https://orcid.org/0000-0003-1032-9558
Ann M Moormann ⬡ https://orcid.org/0000-0003-1113-2829

## Ethics

Written informed consent was obtained from each adult participant and children guardian. Ethical approvals were obtained from the Scientific and Ethics Review Unit (SERU) at the Kenya Medical Research Institute (KEMRI) reference number 3542, and the Institutional Review Board at the University of Massachusetts Chan Medical School, Worcester, MA, USA, IRB number H00014522. Brown University, Providence, RI, USA signed a reliance agreement with KEMRI.

Reviewer #1 (Public review): https://doi.org/10.7554/eLife.98462.3.sa1
Reviewer #3 (Public review): https://doi.org/10.7554/eLife.98462.3.sa2
Reviewer #4 (Public review): https://doi.org/10.7554/eLife.98462.3.sa3
Author response https://doi.org/10.7554/eLife.98462.3.sa4

# Additional files

## Supplementary files
MDAR checklist

## Data availability

Deidentified raw data (flow cytometry, serology) from this manuscript are available from ImmPort platform under the accession study number SDY2534.

The following dataset was generated:

| Author(s) | Year | Dataset title | Dataset URL | Database and Identifier |
|-----------|------|---------------|-------------|-------------------------|
| Forconi C, Kurtis J, Moormann A | 2024 | Human TFH responses to malaria specific antigen using a Kenyan cross sectional study | https:/doi.org/10.21430/M3A4ZTNPYI | ImmPort, 10.21430/M3A4ZTNPYI |

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
