## [Editor Report · eLife Assessment]

This descriptive study used multiparameter spectral flow cytometry and clustering analysis of a subset of CD4 T cells, termed circulating T follicular helper (cTfh), responding to *Plasmodium falciparum* antigens, PfSEA -1A and PfGARP. The results from this comprehensive study provide **valuable** information regarding differences in cTfh response profiles between children and adults living in malaria-endemic Kenya and thus offer a potential usefulness towards improving choices of antigen candidates for malaria vaccines. However, the analysis and interpretation of antigen-specific CD4 cTfh responses remain **incomplete**.

---

## [Referee Report · Reviewer #1 (Public review)]

Summary:

This study aims to understand the malaria antigen-specific cTfh profile of children and adults living in malaria holoendemic area. PBMC samples from children and adults were unstimulated or stimulated with PfSEA-1A or PfGARP in vitro for 6h and analysed by a cTfh-focused panel. Unsupervised clustering and analysis on cTfh was performed. The main conclusions are: (A) the children cohort has a more diverse (cTfh1/2/17) recall responses compared to adults (mainly cTfh17) and, (B) Pf-GARP stimulates better cTfh17 responses in adults, thus a promising vaccine candidate.

Strengths:

This study is, in general, well-designed and with excellent data analysis. The use of unsupervised clustering is a nice attempt to understand the heterogeneity of cTfh cells.

Weaknesses:

The authors have provided additional data in Supplementary Figures 14-16. However, I remain concerned about whether cTfh cells are truly responding to antigen stimulation. In Supplementary Figure 15A-F, the IFNg responses appear as expected, SEB elicits the strongest response, as it stimulates bulk T cells, and the staining is promising, showing a clear distinction between IFNg+ and IFNg- populations. However, in Supplementary Figure 15I-N, the IL-21 secretion assay is concerning. The FACS plots make it difficult to distinguish IL-21+ from IL-21- cells, raising concerns about the validity of this analysis. Additionally, in panel J, the responses to PfSEA-1A or PfGARP appear even greater than those to SEB stimulation. In PBMCs, only a small percentage of T cells should be specific to a particular antigen. How can the positive control (SEB) produce a weaker response than stimulation with a specific antigen? This suggests that the IL-21 secretion assay may not have worked, making the authors' interpretation unreliable.

I also have similar concerns about the IL-4 secretion in Sup Figure 16. First, the FACS plot shows that appear double-positive for IL-21 and IL-4, so it suggests the staining may be due to autofluorescence rather than true cytokine signals. Also in B-C the responses of SEB stimulation is generally weaker than stimulated by one antigen, further questioning the reliability of the IL-4 assay. In summary, I am not convinced that the in vitro antigen stimulation assay worked as intended. Consequently, the manuscript's claims regarding PfSEA-1A- and PfGARP-specific cTfh responses are not sufficiently supported by the presented data.

---

## [Referee Report · Reviewer #3 (Public review)]

Summary:

The goal of this study was to carry out an in-depth granular and unbiased phenotyping of peripheral blood circulating Tfh specific to two malaria vaccine candidates, PfSEA-1A and PfGARP, and correlate these with age (children vs adults) and protection from malaria (antibody titers against Plasmodium antigens.) Authors further attempted to identify any specific differences of the Tfh responses to these two distinct malaria antigens.

Strengths:

The authors had access to peripheral blood samples from children and adults living in a malaria-endemic region of Kenya. The authors studied these samples using in vitro restimulation in the presence of specific malaria antigens. Authors generated a very rich data set from these valuable samples using cutting-edge spectral flow cytometry and a 21-plex panel that included a variety of surface markers, cytokines and transcription factors.

Update following first revision (R1) of the manuscript:

The authors have made a great effort to comprehensively address comments raised by the reviewers. In particular, clearly showing expression of ICOS and Bcl6 on CXCR5+ cells greatly strengthens the case for defining these cells as Tfh-like circulatory lymphocytes (cTfh).

Weaknesses:

Update following first revision (R1) of the manuscript:

Unfortunately, my main concern remains. As it stands, the study is not really on antigen-specific T cells, but rather on the overall CD4 T cell compartment plus or minus antigenic stimulation. Although authors used an in vitro restimulation strategy with malaria antigens, they do not focus on cells de-novo expressing activation markers as a result of restimulation, neither they use tetramers to detect antigen-specific T cells. Moreover, their data shows that the number of CXCR5+ CD4 T cells de-novo expressing activation markers and/or cytokines as a result of their in vitro restimulation is negligible, even when using a prototypic superantigen (SEB).

Thus, no antigen-specific CXCR5+ CD4 T cells could be analysed with the data that the authors provide in this manuscript.

---

## [Referee Report · Reviewer #4 (Public review)]

Summary:

This manuscript is a descriptive study of circulating T follicular helper (cTfh) responses to PfSEA -1A or PfGARP (targets of new antimalaria vaccine candidates) in PBMCs from a convenience sample of children (7 yrs of age) and adults living in a malaria holo endemic Kenya using multiparameter flow cytometry and clustering analysis. This cell type promotes B cell production of long-lived antimalarial antibodies to provide protection against malaria. They find that children had a wider cTFH cytokine and TF profile cellular response in comparison to adults who responded to both antigens but had a narrower response profile.

Strengths:

Carefully done study, very detailed, nice summary model at the end of the paper. The revision provides requested clarification on a number of issues, including CD40L expression which was not differentially expressed between groups. They add additional data into the supplemental files, including IL4 and IL21 data by presenting the cytoplots.

Weaknesses:

To know the significance of these cTfh cells for long-term protection of malaria requires functional and transfer experiments in animal models which is outside the scope of this work.

---

## [Author Response]

The following is the authors’ response to the original reviews

**Public Reviews:**

**Reviewer #1 (Public Review):**
Summary:This study aims to understand the malaria antigen-specific cTfh profile of children and adults living in a malaria holoendemic area. PBMC samples from children and adults were unstimulated or stimulated with *Pf*SEA-1A or *Pf*GARP in vitro for 6h and analysed by a cTfh-focused panel. Unsupervised clustering and analysis on cTfh were performed.The main conclusions are:(1) the cohort of children has more diverse (cTfh1/2/17) recall responses compared to the cohort of adults (mainly cTfh17) and(2) Pf-GARP stimulates better cTfh17 responses in adults, thus a promising vaccine candidate.Strengths:This study is in general well-designed and with excellent data analysis. The use of unsupervised clustering is a nice attempt to understand the heterogeneity of cTfh cells. Figure 9 is a beautiful summary of the findings.Weaknesses:(1) Most of my concerns are related to using *Pf*SEA-1A and *Pf*GARP to analyse cTfh in vitro stimulation response. In vitro, stimulation on cTfh cells has been frequently used (e.g. Dan et al, PMID: 27342848), usually by antigen stimulation for 9h and analysed CD69/CD40L expression, or 18h and CD25/OX40. However, the authors use a different strategy that has not been validated to analyse in vitro stimulated cTfh. Also, they excluded CD25+ cells which might be activated cTfh. I am concerned about whether the conclusions based on these results are reliable.It has been shown that cTfh cells can hardly produce cytokines by Dan et al. However, in this paper, the authors report the significant secretion of IL-4 and IFNg on some cTfh clusters after 6h stimulation. If the stimulation is antigen-specific through TCR, why cTfh1 cells upregulate IL-4 but not IFNg in Figure 6? I believe including the representative FACS plots of IL-4, IFNg, IL21 staining, and using %positive rather than MFI can make the conclusion more convincing. Similarly, the author should validate whether TCR stimulation under their system for 6h can induce robust BCL6/cMAF expression in cTfh cells. Moreover, there is no CD40L expression. Does this mean TCR stimulation mediated BCl6/cMAF upregulation and cytokine secretion precede CD40L expression?In summary, I am particularly concerned about the method used to analyse *Pf*SEA-1A and *Pf*GARP-specific cTfh responses because it lacks proper validation. I am unsure if the conclusions related to *Pf*SEA-1A/*Pf*GARP-specific responses are reliable.

An unfortunate reality of these types of complex immunologic studies is that it takes time to optimize a multiparameter flow cytometry panel, run this number of samples, and then conduct the analysis (not to mention the time it takes for a manuscript to be accepted for peer-review). An unexpected delay, frankly, was the COVID-19 pandemic when non-essential research lab activities were put on hold. We designed our panel in 2019 and referred to the “T Follicular Helper Cells” Methods and Protocols book from Springer 2015. Obviously the field of human immunology took a huge leap forward during the pandemic as we sought to characterize components of protective immunity, and as a result there are several new markers we will choose for future studies of Tfh subsets. We agree with the reviewer that cytokine expression kinetics differ depending on the in vitro stimulation conditions. Due to small blood volumes obtained from healthy children, we were limited in the number of timepoints we could test. However, since we were most interested in IL21 expression, we found 6 hrs to be the best in combination with the other markers of interest during our optimization experiments. We did find IFNg expression from non-Tfh cells, therefore we believe our stimulation conditions worked.

Dan et al used stimulated tonsils cells to assess the CXCR5^pos^PD1^pos^CD45RA^neg^ Tfh and CXCR5^neg^ CD45RA^neg^ non-Tfh whereas in our study, we evaluated CXCR5^pos^PD1^pos^CD45RA^neg^ Tfh from PBMCs. Dan et al PBMCs’ work used EBV/CMV or other pathogen product stimuli and only gated on CD25^pos^OX40^pos^ cells which are not the cells we are assessing in our study. This might explain in part the differences in cytokine kinetics, as we evaluated CD25^neg^ PBMCs only. However, we agree that more recent studies focused on CXCR5^pos^PD1^pos^ cells included more Activation-induced marker (AIM) markers, which are missing in our study, inducing a lack of depth in our analysis.

Percentage of positive cells and MFI are complementary data. Indeed, the percentage of positive cells only indicates which cells express the marker of interest without giving a quantitative value of this expression. MFI indicates how much the marker of interest is expressed by cells which is important as it can indicate degree of activation or exhaustion per cell. Meta-cluster analysis is not ideal to assess the percentage of positivity whereas it does provide essential information regarding the intensity of expression. We added supplemental figures 14 (Bcl6 and cMAF), 15 (INFg and IL21) and 16 (IL4 and IL21) where percentage of positive cells were manually gated directly from the total CXCR5^pos^CD4^pos^CD45RA^neg^CD25^neg^ TfH based on the FMO or negative control, and we overlaid the positive cells on the UMAP of all the CXCR5^pos^CD4^pos^CD45RA^neg^CD25^neg^ meta-clusters. Results from the manual gating are consistent with the results we show using clustering. However, it helps to better visualize that antigen-specific IL21 expression was statistically significant in children whereas the high background observed for adults did not reveal higher expression after stimulation, perhaps suggesting an upper threshold of cytokine expression (supplemental figure 15). The following sentence has been added in the methods at the end of the “OMIQ analysis” section: “ However, the percentage of positive IFN𝛾, IL-4, IL-21, Bcl6, or cMAF using manual gating can be found in Supplemental Figures 14, 15, and 16 along with the overlay of the gated positive cells on the CD4^pos^CXCR5^pos^CD25^neg^ UMAP and the cytoplots of the gated positive cells for each meta-cluster (Supplemental Figures 14, 15, and 16).”

Indeed cMAF can be induced by TCR signaling, ICOS and IL6 (Imbratta et. al, 2020). However, in our study populations, ICOS was expressed (see Author response image 1, panel A) in absence of any stimulation suggesting that CXCR5^pos^CD4^pos^CD25^neg^CD45RA^neg^ cells were already capable of expressing cMAF. Indeed, after gating Bcl6 and cMAF positive cells based on their FMOs (Author response image 1, panel B and C, respectively), we overlaid positive cells on the CXCR5^pos^CD4^pos^CD25^neg^CD45RA^neg^ cells UMAP and we can see that most of our cells already express cMAF alone (Author response image 1, panel D), co-express cMAF and Bcl6 (Author response image 1, panel E), confirming that they are TfH cells, whereas very few cells only expressed Bcl6 alone (Author response image 1, panel F). Because we knew that cT_FH_ already expresses Bcl6 and cMAF, we focused our analysis on the intensity of their expression to assess if our vaccine candidates were inducing more expression of these transcription factors.

**Author response image 1. sa4fig1:** 

(2) The section between lines 246-269 is confusing. Line 249, comparing the abundance after antigen stimulation is improper because 6h stimulation (under Golgi stop) should not induce cell division. I think the major conclusions are contained in Figure 5e, that (A) antigen stimulation will not alter cell number in each cluster and (B) children have more MC03, 06 and fewer MC02, etc. The authors should consider removing statements between lines 255-259 because the trends are the same regardless of stimulations.

We agree, there is no cell division after 6h and that different meta clusters did not proliferate after this short of in vitro stimulation. The use of the word ‘abundance’ in the context of cluster analysis is in reference to comparing the contribution of events by each group to the concatenated data. After the meta clusters are defined and then deconvoluted by study group, certain meta clusters could be more abundant in one group compared to another - meaning they contributed more events to a particular metacluster.

Dimensionality reduction is more nuanced than manual gating and reveals a continuum of marker expression between the cell subsets, as there is no hard “straight line” threshold, as observed when using in 2D gating. Because of this, differences are revealed in marker expression levels after stimulation making them shift from one cluster to another - thereby changing their abundance.

To clarify how this type of analysis is interpreted, we have modified lines 255-259 as follows:

“In contrast, the quiescent *Pf*SEA-1A- and *Pf*GARP-specific cT_FH_2-like cluster (MC02) was significantly more abundant in adults compared to children (Figure 5c and 5d, *pf*<0.05). Interestingly, following *Pf*GARP stimulation, the activated cT_FH_1/17-like subset (MC09) became more abundant in children compared to adults (Figure 5d, *pf*<0.05 with a False Discovery Rate=0.08), but no additional subsets shifted phenotype after *Pf*SEA-1A stimulation (Figure 5c).”

**Reviewer #2 (Public Review):**
Summary:Forconi et al explore the heterogeneity of circulating Tfh cell responses in children and adults from malaria-endemic Kenya, and further compare such differences following stimulation with two malaria antigens. In particular, the authors also raised an important consideration for the study of Tfh cells in general, which is the hidden diversity that may exist within the current 'standard' gating strategies for these cells. The utility of multiparametric flow cytometry as well as unbiased clustering analysis provides a potentially potent methodology for exploring this hidden depth. However, the current state of analysis presented does not aid the understanding of this heterogeneity. This main goal of the study could hopefully be achieved by putting all the parameters used in one context, before dissecting such differences into their specific clinical contexts.Strengths:Understanding the full heterogeneity of Tfh cells in the context of infection is an important topic of interest to the community. The study included clinical groupings such as age group differences and differences in response to different malaria antigens to further highlight context-dependent heterogeneity, which offers new knowledge to the field. However, improvements in data analyses and presentation strategies should be made in order to fully utilize the potential of this study.Weaknesses:In general, most studies using multiparameter analysis coupled with an unbiased grouping/clustering approach aim to describe differences between all the parameters used for defining groupings, prior to exploring differences between these groupings in specific contexts. However, the authors have opted to separate these into sections using "subset chemokine markers", "surface activation markers" and then "cytokine responses", yet nuances within all three of these major groups were taken into account when defining the various Tfh identities. Thus, it would make sense to show how all of these parameters are associated with one another within one specific context to first logically establish to the readers how can we better define Tfh heterogeneity. When presented this way, some of the identities such as those that are less clear such as "MC03/MC04/ MC05/ MC08" may even be better revealed. once established, all of these clusters can then be subsequently explored in further detail to understand cluster-specific differences in children vs adults, and in the various stimulation conditions. Since the authors also showed that many of the activation markers were not significantly altered post-stimulation thus there is no real obstacle for merging the entire dataset for the first part of this study which is to define Tfh heterogeneity in an unbiased manner regardless of age groups or stimulation conditions. Other studies using similar approaches such as Mathew et al 2020 (doi: 10.1126/science.abc8) or Orecchioni et al 2017 (doi: 10.1038/s41467-017-01015-3) can be referred to for more effective data presentation strategies.Accordingly, the expression of cytokines and transcription factors can only be reliably detected following stimulation. However, the underlying background responses need to be taken into account for understanding "true" positive signals. The only raw data for this was shown in the form of a heatmap where no proper ordering was given to ensure that readers can easily interpret the expression of these markers following stimulation relative to no stimulation. Thus, it is difficult to reliably interpret any real differences reported without this. Finally, the authors report differences in either cluster abundance or cluster-specific cytokine/ transcription factor expression in Tfh cell subsets when comparing children vs adults, and between the two malaria antigens. The comparisons of cytokine/transcription factor between groups will be more clearly highlighted by appropriately combining groupings rather than keeping them separate as in Figures 6 and 7.

Thank you for sharing these references. Similar to SPADE clustering and ViSNE dimensionality algorithms used in Orecchioni et al, we used all the extracellular markers from our panel in our FlowSOM algorithm with consensus meta-clustering which includes both the chemokine receptors and activation markers even though they are presented separately in our manuscript across the figure 3 and 4. This was explained in the methods section (lines 573 - 587). We then chose the UMAP algorithm as visual dimensionality reduction of the meta-clusters generated by FlowSOM-consensus meta-clustering as explained under the “OMIQ analysis” subpart of our methods (lines 588- 604). Therefore, we believe we have conducted the analysis as this reviewer suggests even if we chose to show the figures that were informative to our story. The heatmap of the results brings the possibility to see which combination of markers respond or not to the different conditions and between groups, all the raw data are present from the supplemental figures 10 to 13 showing, using bar plots, the differences expressed in the heatmaps. We believe it strengthens our interpretation of the results.

Regarding the transcription factor and cytokine background, we added supplemental figures 14, 15 and 16 where we used manual gating to select Bcl6, cMAF, IFNg, IL21 or IL4 positive cells directly from total CXCR5^pos^CD4^pos^CD45RA^neg^CD25^neg^ TfH cells based on the FMO or negative control, and we overlaid the positive cells on the UMAP of all the CXCR5^pos^CD4^pos^CD45RA^neg^CD25^neg^ meta-clusters. Moreover, all the dot plots (with their statistics) used for the heatmap figure 6 and 7 can be found in the supplemental figures 10, 11, 12 and 13. These supplemental figures address the concerns above by showing the difference of signals between unstimulated and stimulated conditions.

**Reviewer #3 (Public Review):**
Summary:The goal of this study was to carry out an in-depth granular and unbiased phenotyping of peripheral blood circulating Tfh specific to two malaria vaccine candidates, *Pf*SEA-1A and *Pf*GARP, and correlate these with age (children vs adults) and protection from malaria (antibody titers against Plasmodium antigens.). The authors further attempted to identify any specific differences in the Tfh responses to these two distinct malaria antigens.Strengths:The authors had access to peripheral blood samples from children and adults living in a malaria-endemic region of Kenya. The authors studied these samples using in vitro restimulation in the presence of specific malaria antigens. The authors generated a very rich data set from these valuable samples using cutting-edge spectral flow cytometry and a 21-plex panel that included a variety of surface markers, cytokines, and transcription factors.Weaknesses:- Quantifying antigen-specific T cells by flow cytometry requires the use of either 1- tetramers or 2- in vitro restimulation with specific antigens followed by identification of TCR-activated cells based on de-novo expression of activation markers (e.g. intracellular cytokine staining and/or surface marker staining). Although authors use an in vitro restimulation strategy, they do not focus their study on cells de-novo expressing activation markers as a result of restimulation; therefore, their study is not really on antigen-specific cTfh. Moreover, the authors report no changes in the expression of activation markers commonly used to identify antigen-specific T cells upon in vitro restimulation (including IFNg and CD40L); therefore, it is not clear if their in vitro restimulation with malaria antigens actually worked.

We understand the reviewer’s point of view and apologies for any confusion. IFNg was expressed but not statistically different between groups. Indeed, looking at the CD8 T cells and using manual gating, we were able to show that IFNg was increased but not statistically significant upon stimulation from CD4^pos^CXCR5^pos^ cells (supplemental figure 15, panel C), confirming our primary observation using clustering analysis. These results showed that our malaria antigen induced IFNg response in some participants, but not all of them, revealing heterogeneity in this response among individuals within the same group.

Regarding CD40L, in the supplemental figure 7, we can see that some of our meta-clusters expressed more CD40L upon stimulation, but again without leading to statistical differences between groups. Combined with the increased expression of other cytokines and transcription factors, we showed that our stimulation did indeed work. However, because of the high variation within groups, there were no statistical differences across our groups. Because CD40L is not the only marker showing specific T cell activation, and not all T cells respond using this marker alone, a more comprehensive multimarker AIM panel might have highlighted differences between groups. We recognized the limitations of our study and believe that future study will benefit from more activation markers commonly used to identify antigone-specific T cells such as CD69, OX40, 4-1BB (AIM panel), among other markers.

- CXCR5+CD4+ memory T cells have been shown to present multi-potency and plasticity, capable of differentiating to non-Tfh subsets upon re-challenge. Although authors included in their flow panel a good number of markers commonly used in combination to identify Tfh (CXCR5, PD-1, ICOS, Bcl-6, IL-21), they only used one single marker (CXCR5) as their basis to define Tfh, thus providing a weak definition for Tfh cells and follow up downstream analysis.

Sorry for the confusion, even though the subsampled on the CD4^pos^CXCR5^pos^ CD25^neg^ cells to run our FlowSOM, we showed the different levels of expression across meta-clusters (figure 4 panels A and B) of PD1 (Tfh being PD1 positive cells) and ICOS (indicating the activation stage of the Tfh, “T Follicular Helper Cells” Methods and Protocols book from Springer 2015). We also included an overlay of the manually gated double positive Bcl6-cMAF cells on the CXCR5^pos^CD45RA^neg^CD25^neg^ CD4 T cell UMAP plot to show that most of them express Bcl6 (supplemental figure 14). Interestingly, the manually gated IL21 positive cells were less abundant, particularly for children (supplemental figure 15). Because we were not able to include all the markers that are now used to define Tfh cells, we referred to our cell subsets as “TFH-like”. This is an acknowledged limitation of our study. Due to the limited blood volume obtained from children and cost of running multiplex flow cytometry assays, our results showing antigen-specific heterogeneity of Tfh subset will have to be validated in future studies that include these additional defining markers.

- Previous works have used FACS-sorting and in vitro assays for cytokine production and B cell help to study the functional capacity of different cTfh subsets in blood from Plasmodium-infected individuals. In this study, authors do not carry out any such assays to isolate and evaluate the functional capacity of the different Tfh subsets identified. Thus, all the suggestions for the role that these different cTfh subsets may have in vivo in the context of malaria remain highly hypothetical.

Unfortunately, low blood volumes obtained from children prevented us from running in vitro functional assays and the study design did not allow us to correlate them with protection. However, since the function of identified Tfh subsets from malaria-exposed individuals has been evaluated using Pf lysates in other studies, we referenced them when interpreting the differences we reported in Tfh subset recognition between malaria antigens. If either of these antigens move forward into vaccine trials, then evaluating their function would be important.

- The authors have not included malaria unexposed control groups in their study, and experimental groups are relatively small (n=13).

This study design did not include the recruitment of malaria naive negative controls as its goal was to assess malaria antigen-specific responses comparing the quality and abundance between malaria-exposed children to adults to these potential new vaccine targets *Pf*SEA-1A and *Pf*GARP. We did however test 3 malaria-naive adults and found no non-specific activation after stimulation with these two malaria antigens. Since this was done as part of our assay optimization, we did not feel the need to show these negative findings.

And even with our small sample size, we demonstrated significant age-associated differences in malaria antigen-specific responses from cT_FH_-like subsets.

**Recommendations for the authors:**

**Reviewer #1 (Recommendations For The Authors):**
Minor points are:(1) Line 88, cTfh cells are not only from GC-Tfh, they have GC-independent origin (He et al, PMID: 24138884).

The following sentence was added line 88 “Interestingly, cT_FH_ cells can also come from peripheral cT_FH_ precursor CCR7^low^PD1^high^CXCR5^pos^ cells; thus, they also have a GC-independent origin (He, Cell, 2013 PMID: 24138884).

(2) I believe all participants were free of blood-stage infection upon enrolment. But can authors clearly state this information between lines 151-159?

We mentioned in the methods, line 495-496 “Participants were eligible if they were healthy and not experiencing any symptoms of malaria at the time venous blood was collected”. However, using qPCR we found 5 children with malaria blood stage. As shown in Author response image 2, comparing malaria free to blood-stage children, no differences were observed without any stimulation. However, MC03 is more abundant upon malaria antigen stimulation in the blood-stage group whereas MC04 is more abundant in the malaria free group upon *Pf*GARP stimulation only confirming that our stimulation worked.

**Author response image 2. sa4fig2:** 

**Reviewer #3 (Recommendations For The Authors):**
(1) The strategy for gating on antigen-specific cTfh cells needs to be revised. The correct approach would be to gate on those cells that respond by de-novo expression of activation markers upon antigen restimulation (also termed activation-induced markers. e.g. CD69, CD40L, CXCL13 and IL-21, Niessl 2020; CD69, CD40L, CD137 and OX40, Lemieux 2023; CD137 and OX40, Grifoni 2020). As it stands, the study is not really on antigen-specific T cells, but rather on the overall CD4 T cell compartment plus or minus antigenic stimulation.

We recognized the limitation in our flow panel design which prevents us from performing this gating. We originally based our panel design on the “T follicular helper cells methods and protocols” book (Springer 2015) which used CD45RA, CD25, CXCR5, CCR6, CXCR3, CCR7, ICOS and PD1 to define cT_FH_. We had already optimized our 21-color panel, purchased reagents and started to run our experiments by the time these publications modified how to define TFH cells Niessl, Lemieux and Grifoni’s publication. Indeed we optimized and performed our assay from November 2019 to March 2020, finishing to run the samples during the first quarantine. Because of the urgent needs of research on SARS-CoV-2 that we were involved with from this time and moving forward, the analysis of our TFH work got highly postponed. Moreover, 2020 is also the year where many TFH papers came out with better ways to define cT_FH_ and responses to antigen stimulations. In our future studies, our panel will include AIM.

(2) It is not clear if the antigenic stimulation actually worked. Does the proportion of IFNg+ or IL-4+ or IL-21+ or CD40L+ or CD25+ CD4 or CD8 T cells increase following in vitro antigen restimulation?

Yes, using manual gating, we are able to show an increase of IL4 (supplemental figure 16 panel B and C), and IL21 (supplemental figure 15 panel J and K) production in both children and adults. However, we did not observe significant production of IFNg (supplemental figure 15, panel C) and changes in CD40L expression (supplemental figure 7) after malaria antigen stimulation, however, our positive control SEB worked. So, yes our stimulation assay worked but these 2 malaria antigens did not significantly induce these cytokines. This could be that they are too low to detect in every participant since they are single antigens and not whole parasite lysates, as other studies have used. It could also be that these antigens don’t stimulate CD40L or IFNg in all our participants. We brought up this limitation as follow in the discussion, line 473: “Although the heterogeneity in the response of CD40L and IFNγ suggests that our tested malaria antigens did not induce significant differences in the expression of these markers in all our participants, our panel did not include other activated induced markers, such as OX40, 4-1BB, and CD69”.

(3) It is not clear what is the proportion of cTfh over the total CD4 T cell compartment among the different groups. Does this vary among different groups? It would be valuable to display this as an old-fashioned combination of contour plots with outliers for illustrating flow cytometry and bar graphs for the cumulative data.

The proportion of CD3^pos^CD4^pos^CD25^neg^CXCR5^pos^ cTfh cells did not differ within the total number of CD4 T cells between groups (figure 2).

(4) The gating strategy could be refined and become more robust if adding additional markers in combination with CXCR5 for identifying cTfh (e.g. CXCR5+Bcl6+).

Thank you for this suggestion. An overlay of Bcl6 expression can be found in supplemental figure 14 where we confirm that our CXCR5+ cT_FH_-like subsets express cMAF and Bcl6.

(5) The protocols for intracellular and intranuclear staining seem to be incomplete in Materials and Methods. In particular, cell permeabilization strategies seem to be missing.

Our apologies for this oversight, we added the following sentences in the methods line 545: “Cells were fixed and permeabilized for 45 mins using the transcription factor buffer set (BD Pharmingen) followed by a wash with the perm-wash buffer. Intracellular staining was performed at 4 °C for 45 more mins followed by two washes using the kit’s perm-wash buffer”.

(6) In Materials and Methods, the authors mention they have used fluorescence minus one control to set their gating strategy. It would be valuable to show these, either on the main body or as part of supplementary figures.

We added the cytoplots of the FMOs and/or negative controls as appropriate in the supplemental figures 14 (cMAF and Bcl6), 15 (IFNg and IL21) and 16 (IL4 and IL21).

(7) Line 194 and Figure 3, it is not clear the criteria that the authors used for down-sampling events before FlowSOM analysis. Was this random? Was this done with unstimulated or stimulated samples?

We chose to down-sample on CD3posCD4^pos^CD25^neg^CD45RA^neg^ and CXCR5^pos^ cells prior to our FlowSOM to allow more cluster analysis to focus only on the differences among those cells. The down-sampling used 1,000 CD3posCD4^pos^CD25^neg^ CD45RA^neg^CXCR5^pos^ cells from each fcs file (unstimulated and stimulated samples). If the fcs file had more than 1,000 CXCR5^pos^ cells, the down-sampling was done randomly by the OMIQ platform algorithm to select only 1,000 CXCR5^pos^ cells within this specific fcs file. The latest sentence was added to the methods line 593.

(8) Lanes 201, 202, As it stands, the take of the authors on the role of different cTfh subsets during infection remains highly speculative. Are these differences in cTfh phenotypes actually reflected in their in vitro capacity to provide B cell help (e.g. as in the Obeng-Adjei 2015 paper) or to produce IL-21, express co-stimulatory molecules, or any other characteristic that would allow them to better infer their functional roles during infection? Any additional in vitro analysis of the functional capacity of isolated cTfh subsets identified in this research would greatly increase its value.

We agree with the reviewer that this sentence is speculative, and we rephrase it as follow: “First, we found different CXCR5 expression levels between meta-clusters (Figure 3b); CXCR5 is essential for cT_FH_ cells to migrate to the lymph nodes and interact with B-cells”. We would have liked to perform in vitro functional assays. However, as explained above, we did not have sufficient cells collected from children to do so.

(9) It is not clear why authors omitted IL-17 and did not use IFNg and IL-4 to refine their definition of Th1, Th2 and Th17 cTfh.

We would have liked to include IL-17, however we were constrained by only having access to a 4 lasers cytometer at the time we ran our assay. In light of needing to prioritize markers, when we were designing our flow panel, cTfh1 were shown to be preferentially activated during episodes of acute febrile malaria children (Obeng-Adjei). Therefore, we chose to focus on IFNg and IL4 to differentiate Tfh1 from Tfh2, in addition to other markers as surrogate of functional potential. We did not use IFNg and IL4 to refine our definition of Tfh1, Tfh2 and Tfh17 as recent publications have shown that IL4 is not only expressed in Tfh2 but also in the other Tfh subsets, at lower intensity (Gowthaman among others). Therefore IFNg and IL4 by themselves were not sufficient to properly define the different Tfh subsets. In future studies, we plan to include transcription factor profiles (T-bet, BATF, GATA3) to further refine definitions of Tfh subsets.

(10) Lines, 226, 228, based on the combination of markers that the MC03 subset expresses, it is tempting to think that this is the only "truly" committed Tfh subset from the entire analysis. Please, discuss.

If the reviewer is referring to changes in marker expression levels that indicate they have not reached a level of differentiation that would make them reliable (ie “true) Tfh cells, we agree that this is an important question now that we have technology that can measure and analyse so many phenotypic markers at once. This brings forward the need for the scientific method - to replicate study findings to determine whether they are consistent given the same study design and experimental conditions.

(11) Lines 243 244, Again, is this reflected in functional capacity?

The study described in this manuscript did not include functional assays. However, this did not change the key finding that different malaria antigens behaved differently, demonstrating heterogeneity in Tfh recognition of malaria antigens. Regarding CD40L expression, we did not observe differences between groups, however some individuals had an increase of their CD40L (supplemental figure 7). It is possible that some individuals had responded through other activated induced markers (CD69, ICOS, OX40, 4-1BB among others) and that our stimulation condition was not long enough to assess CD40L expression upon malaria antigen stimulation. This limitation has been addressed by editing the line 243-244 as follows: “we were unable to find statistical differences in the CD40L expression between groups as only few individuals responded through it (supplemental figure 7).”

(12) Lines 243, 244, Are these cTfh subsets exclusively detected in malaria-exposed individuals? This is confounded by the lack of a malaria unexposed control group in this study, which would have been highly valuable.

We agree with the reviewer that having non-naive children would have been valuable as a negative control group. However, this study was conducted in Kenya where all children are suspected to have had at least one malaria infection. We also did not have ethical approval or the means to enroll children in the USA who would not have been exposed to malaria as a negative control group. Since we were also evaluating differences by age group, comparing US adults would not have helped to address this point. Therefore, this remains an open question that might be addressed by another study recruiting children in non-malaria endemic areas.

(13) Line 267, as the authors have not gated on T cells de-novo expressing activation markers in response to antigen restimulation, how do they know these are indeed antigen-specific cTfh?

Omiq analysis accounts for marker expression levels in the resting cells (unstimulated well) for each individual compared to each experimental/stimulated well. The algorithm computationally determines whether that expression level changed without an arbitrary positive threshold, keeping the expression levels as a continuous variable, not dichotomous - which is the power of unbiased cluster analyses. Therefore, we know that these cells are antigen-specific based on the statistical difference in intensity expression between the resting cells and the stimulated ones. Nevertheless, manual gating to show “de-novo” responding cells, produced the same results as assessing the MFI of each meta-cluster (supplemental figures 14, 15 and 16).

(14) Lines, 292-295, it is very surprising that Tfh cells would not produce IL-21 upon restimulation. Have the authors observed upregulation of IL-21 following SEB restimulation?

Yes, we observed IL21 positive cells upon SEB stimulation (supplemental figure 15, panel J and K). However we found unexpectedly high background levels of IL21, specifically within the adult group (supplemental figure 15, panel K and M) making it challenging to find antigen-specific increases above background. Interestingly, an increase in IL21 using manual gating was observed upon *Pf*SEA-1A or *Pf*GARP stimulation in children (supplemental figure 15, panel J and L).

(15) In Figures 3 and 4, it is not clear if there are any significant differences in expression of different markers between different cTfh subsets and/or different conditions. Moreover, the lack of differences in response to antigen stimulation seems to suggest that it did not work adequately.

We intentionally chose 6-hours stimulation to better assess changes in cytokines which we did. However, because it is a short stimulation, we did not expect dramatic changes in the extracellular markers presented in the figure 3 and 4. A longer stimulation, such as 24h, will highlight properly these changes.

(16) Figure 5b would benefit from bar graphs.

Please find below the bar-graphs for the highlighted meta-clusters in figure 5b. We did not include these bar-graphs to our figure 5 as they do not bring new information. They repeat the information already presented through the EdgeR plot.

**Author response image 3. sa4fig3:** 

(17) Figures 6 and 7 would greatly benefit from showing individual examples of old-fashioned contour with outliers flow plots to illustrate the different cTfh subsets identified in the study.

The different cT_FH_ subsets can be found with a contour plot with outliers in the supplemental figure 4.

(18) Figures 3,4, 6, and 7, the authors exclusively focused on the study of MFI to measure the expression of cytokine and transcription factors among different groups/stimulations. Have the authors observed any differences in the percentage or absolute counts of cytokine+ and/or TF+ between different subsets of cTfh and/or different conditions?

Yes. We added the supplemental figures 14 (transcription factors) and 15/16 (cytokines) where cytokines and transcription factors were assessed using manual gating. We found that total CD4^pos^CXCR5^pos^ IL4 was significantly increased upon stimulation in both adults and children while IFNg was not. However, we found significantly higher IFNg on total CD8^pos^ cells showing that the stimulation worked, but the total CD4^pos^CXCR5^pos^ did not express IFNg. Finally, we observed a trend of higher IL21^pos^CD4^pos^CXCR5^pos^ in adults, not significant due to high background whereas IL21 was significantly increased upon stimulation in children. Regarding cMAF and Bcl6, both transcription factors were significantly increased upon stimulation within children only.

(19) Figure 8, the definition for high and low *Pf*GARP antibody titers seems rather arbitrary. Are these associations still significant when attempting a regular correlation analysis between Ab values (i.e. Net MFI) and different cTfh subsets?

Yes, the definition for high and low *Pf*GARP antibody levels is arbitrary but when looking at the antibody data (figure 1b), it was naturally bimodal. Therefore as a sub-analysis, we assess the association between *Pf*GARP antibodies levels and cT_FH_ subsets, see Author response image 4. We checked the correlation between the abundance of the meta-clusters and the level of IgG anti-*Pf*GARP and anti-*Pf*SEA after *Pf*GARP and *Pf*SEA stimulation. We also checked the correlation between the MFI expression of Bcl6 and cMAF after stimulation (*Pf*GARP or *Pf*SEA-1A minus the unstimulated) by the meta-clusters and the level of IgG anti-*Pf*GARP and anti-*Pf*SEA. However, we believe that because of our small sample size, our results are not robust enough and that we risk over-interpreting the data. Therefore, we choose not to include this analysis in the manuscript.

**Author response image 4. sa4fig4:** 

(20) The comprehensive 21-plex panel that authors used in this study could generate insights on additional immune cells beyond cTfh (e.g. additional CD4 T cell subsets, CD8 T cells, CD19 B cells). It is not clear why the authors limited their analysis to cTfh only.

The primary goal of the study was to assess the cT_FH_ response to malaria vaccine candidates. However, we were able to assess the IFNg expression for CD8 T cells upon stimulation using the manual gating as indicated in the supplemental figure 15. Without additional markers to more clearly define other CD4 T cell or B cell subsets, we do not believe this dataset would go deep enough into characterizing antigen-specific responses to malaria antigens that would yield new insight.

(21) Minor point, the punctuation should be revised throughout the manuscript.

Punctuation was revised throughout the manuscript by our departmental scientific writer Dr. Trombly, as per reviewer request.